# SLɪM: One-shot Quantization and Sparsity with Low-rank Approximation for LLM Weight Compression

**Mohammad Mozaffari** [1]   **Amir Yazdanbakhsh** [2]   **Maryam Mehri Dehnavi** [1 3]

## Abstract

Conventional model compression techniques for LLMs address high memory consumption and slow inference challenges but typically require computationally expensive retraining to preserve accuracy. In contrast, one-shot compression methods eliminate retraining costs, but struggle to achieve accuracy comparable to dense models. This paper presents SLɪM, a new one-shot compression framework that holistically integrates *hardware-friendly* quantization, sparsity, and low-rank approximation into a unified process. First, we formulate the quantization process using a probabilistic approach (SLɪM-Quant) that enables us to apply uniform quantization. Then, we use an existing one-shot pruning method to apply semi-structured sparsity on top of the quantized weights. Finally, to compensate for the introduced aggregated quantization and sparsity error, we use a novel saliency function with unique invertible and additive features that enables us to mathematically compute the value of low-rank adapters. SLɪM improves model accuracy by up to 5.66% (LLaMA-2-7B) for 2:4 sparsity with 4-bit weight quantization, outperforming prior methods. Models compressed with SLɪM achieve up to $4.3\times$ and $3.8\times$ layer-wise speedup on Nvidia RTX3060 and A100 GPUs, respectively. Additionally, they achieve up to $0.23\times$ end-to-end memory reduction in comparison to their dense counterparts. We also propose an *optional* PEFT recipe that further improves accuracy by up to 1.66% (LLaMA-2-13B) compared to SLɪM without fine-tuning.[1]

## 1. Introduction

LLMs (Team et al., 2024; Dubey et al., 2024) have significantly advanced natural language understanding and generation (Suzgun et al., 2022; Zhou et al., 2023). However, their extensive parameter counts lead to significant memory overhead and high inference costs (Frantar & Alistarh, 2023; Ma et al., 2024; Guo et al., 2024). To mitigate these challenges, recent methods (Shao et al., 2023; Sun et al., 2023; Lin et al., 2024) leverage compression to reduce inference costs while aiming to retain accuracy as much as possible, albeit often with some trade-offs compared to dense models.

Pruning and quantization methods effectively reduce the computational and memory overhead of LLMs, but they often require costly retraining on large-scale datasets to restore accuracy (Sanh et al., 2020; Park et al., 2018). The computational overhead of retraining, coupled with the numerical and optimization challenges of fine-tuning quantized models (Gholami et al., 2022), makes these approaches impractical for many real-world applications (Frantar et al., 2022).

To eliminate the need for retraining, one-shot compression methods have gained traction, achieving high accuracy using only a relatively small set of calibration data. State-of-the-art methods such as SparseGPT (Frantar & Alistarh, 2023) and Wanda (Sun et al., 2023) have demonstrated strong one-shot pruning performance. However, these methods perform well with unstructured sparsity but struggle with semi-structured patterns, such as NVIDIA's 2:4 sparsity pattern (Mishra et al., 2021), which is crucial for efficient hardware-accelerated inference. To solve this challenge, unlike SparseGPT and Wanda that focus on layer-wise error minimization, MaskLLM (Fang et al., 2024) uses a learnable mask that minimizes the end-to-end model error, resulting in a significant boost to the accuracy of the model at the cost of an expensive mask training phase. One-shot quantization methods use various methods including row/column reordering (Frantar & Alistarh, 2023), scaling (Xiao et al., 2023; Lin et al., 2024), and other transformations (Shao et al., 2023; Ma et al., 2024) to mitigate model accuracy loss. However, their dependence on complex GPU kernel implementations can limit the practical benefits of quantization and add challenges to their development.

---

[1]Department of Computer Science, University of Toronto [2]Google DeepMind [3]NVIDIA Research. Correspondence to: Mohammad Mozaffari <mmozaffari@cs.toronto.edu>.

*Proceedings of the $42^{nd}$ International Conference on Machine Learning*, Vancouver, Canada. PMLR 267, 2025. Copyright 2025 by the author(s).

[1]Code and data for SLɪM is available at: https://github.com/Mohammad-Mozaffari/slim

Although pruning and quantization are effective individually in reducing model size and inference costs, combining them provides even greater compression potential (Frantar & Alistarh, 2023). However, jointly applying these techniques often compounds the accuracy degradation introduced by each method, leading to a significant performance gap between compressed and original models. Recent work (Guo et al., 2024) attempts to mitigate this issue by jointly pruning and quantizing weights. While effective at 8-bit precision, this work struggles to recover the accuracy of dense models under lower bit-width quantization. This persistent accuracy gap highlights the need for alternative compression techniques that can maintain efficiency while minimizing quality degradation, especially at lower bit widths.

Low-rank adapters have emerged as a promising approach to mitigate the accuracy loss introduced by model compression techniques. Recent studies (Guo et al., 2023; Li et al., 2023) have explored learnable low-rank adapters to reduce weight reconstruction errors caused by pruning or quantization. However, these methods typically require an expensive retraining process on hundreds of millions of tokens to recover performance (Dettmers et al., 2023; Nikdan et al., 2024). This retraining is necessary because low-rank adapters are often initialized based on weight norms rather than their direct impact on model outputs, resulting in suboptimal starting points (Guo et al., 2023). To address this limitation, $L^2$QER (Zhang et al., 2024a) introduces a one-shot low-rank adapter approach that compensates for quantization loss by directly minimizing its impact on model outputs. QERA (Zhang et al., 2024b) and CALDERA (Saha et al., 2024) find closed form solutions to the problem that $L^2$QER tries to solve. Although effective for quantization, methods such as $L^2$QER struggle to maintain accuracy when combined with sparsity, underscoring the need for compression techniques that seamlessly integrate low-rank approximations with both sparsity and quantization[2]

To address these limitations, we propose SLIM, a one-shot compression framework that seamlessly integrates hardware-friendly sparsity, quantization, and low-rank approximation to minimize accuracy degradation while maintaining computational efficiency. We decompose the primary objective of SLIM–*one-shot hardware-friendly sparsity and quantization with minimal accuracy loss*–into three key sub-tasks. For quantization, we prioritize uniform quantization (e.g., one scale per tensor) because of its computational efficiency on commodity hardware and its simplified encoding/decoding process. However, standard uniform quantization often introduces significant quantization errors, particularly in tensors with a wide dynamic range or outliers (Gholami et al., 2022). These errors degrade model accu-

racy, making uniform quantization less attractive than more complex per-group quantization methods. Optimizing uniform quantization is inherently a non-convex problem, and recent approaches (Nagel et al., 2021) rely on grid search to find an optimal scaling factor. However, grid search can be both suboptimal and computationally expensive, often leading to poor model quality. To address this, we introduce SLIM-Quant, a probabilistic formulation of the quantization process. Our approach reformulates the inherently non-convex quantization problem into a convex optimization problem, allowing for an efficient and tractable solution. This transformation significantly reduces uniform quantization error, achieving accuracy levels comparable to more complex group quantization methods while retaining the computational efficiency of uniform quantization (up to 6% speedup). After quantization, we apply Wanda (Sun et al., 2023), a state-of-the-art pruning method, to introduce different forms of unstructured and structured sparsity on the quantized weights.

While quantization and pruning significantly reduces the model's computational and memory footprint, the resulting compression error is unavoidable, underscoring the need for effective mitigation strategies. To address this, we propose SLIM-LoRA, a one-shot low-rank adaptation method designed to compensate for the aggregated error introduced by quantization and sparsity. However, determining the optimal values for adapter matrices typically requires an expensive retraining process, making it impractical for large-scale models. To eliminate this retraining overhead, we develop a novel saliency function that is both invertible and additive, enabling us to mathematically derive the low-rank adapter values without iterative optimization. These properties allow SLIM-LoRA to effectively correct compression-induced errors while maintaining computational efficiency.

Compared to state-of-the-art methods, SLIM achieves an average accuracy improvement of 5.66% on LLaMA-2-7B, 3.89% on LLaMA-2-13B, and 0.60% on OPT-13B under 2:4 sparsity and 4-bit weight quantization. More importantly, SLIM shifts the Pareto frontier, delivering higher model accuracy at the *same total bit budget* compared to existing compression techniques (up to 0.5%) and even outperforming dense models (up to 0.6%). These results highlight the effectiveness of SLIM in maximizing model quality under stringent resource constraints, making it an appealing solution for efficient large-scale deployment. Beyond accuracy improvements, SLIM achieves up to 4.3× and 3.8× layer-wise speedup on NVIDIA RTX3060 and A100 GPUs, respectively, demonstrating its efficiency on modern hardware. Finally, to further narrow the gap between compressed and dense models, we use an optional lightweight PEFT method, which provides up to an 1.66% additional accuracy improvement for LLaMA-2-13B.

---

[2]For a more detailed discussion of the related work, see Appendix S.

## 2. Preliminaries

**Model Compression.** Model compression reduces the compute and memory demands of large models while maintaining predictive accuracy by minimizing output differences between compressed and original models. However, directly optimizing these differences across the entire model is computationally infeasible due to the high dimensionality of neural networks. Optimal Brain Surgeon (OBS) (Hassibi et al., 1993) simplifies this challenge by focusing on minimizing output discrepancies layer by layer, using calibration datasets.

OBS applies a layer-wise approach to compress feed-forward layers efficiently. Denoting compressed matrices with a superscript $C$, for a layer with input $\mathcal{X} \in \mathbb{R}^{b \times d_{in}}$, weight $\mathcal{W} \in \mathbb{R}^{d_{in} \times d_{out}}$, and output $\mathcal{Y} \in \mathbb{R}^{b \times d_{out}}$, it minimizes output differences by optimizing Equation 1. This method ensures compression fidelity and has become foundational for many modern compression techniques.

$$\min_{\mathcal{W}^C} |\mathcal{Y}^C - \mathcal{Y}|^2 = \min_{\mathcal{W}^C} |\mathcal{X}(\mathcal{W}^C - \mathcal{W})|^2 \quad (1)$$

**Symmetric Quantization.** Symmetric quantization is a core technique for reducing model size and boosting computational efficiency. It computes the quantized matrix $\mathcal{M}^Q \propto round(\frac{\mathcal{M}}{\alpha})$, where $\alpha$ is a scaling factor based on the range or norm of the matrix. This scaling ensures $\mathcal{M}^Q$ values stay within the representable range, enabling efficient matrix multiplications with minimal overhead. However, its effectiveness depends on selecting $\alpha$ carefully, as this choice significantly impacts precision.

AbsMax, the most common symmetric quantization method, selects $\alpha$ as the matrix's maximum absolute value, ensuring all values remain within the target range. Unfortunately, it is highly sensitive to outliers; a single large value can inflate $\alpha$, reducing the precision of most quantized weights. For zero-centered, bell-curved distributions typical in LLMs, AbsMax maps many weights to zero, leading to significant quantization errors.

Group quantization (Alistarh et al., 2017; Gunho et al., 2022) tackles AbsMax's outlier sensitivity by assigning separate scaling factors to subgroups of the weight matrix. This approach captures local variations in weight magnitudes, reducing quantization error for non-uniform distributions. However, storing multiple scaling factors increases memory usage, and subgroup-specific dequantization increase computational complexity, potentially slowing inference. The challenges of using group quantization are discussed in Appendix U.

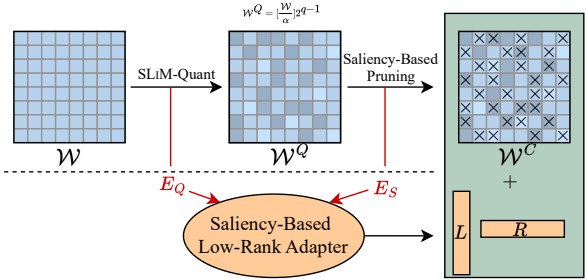

*Figure 1.* The SLIM weight compression pipeline consists of three main steps: (1) Quantizing weights using the symmetric SLIM-Quant algorithm, producing quantized weights $\mathcal{W}^Q$ and quantization error $E_Q$; (2) Sparsifying quantized weights $\mathcal{W}^Q$ through a pruning method, resulting in compressed weights $\mathcal{W}^C$ and sparsity error $E_S$; (3) Mitigating compression errors through SLIM saliency-based low-rank approximation, generating left and right low-rank adapters $L$ and $R$. Optionally, these adapters can be fine-tuned with sparse quantized weights frozen to further enhance model accuracy.

## 3. Quantized Sparse Plus Low-rank Approximation of LLMs

To achieve effective compression of LLMs while preserving accuracy, SLIM combines quantization, pruning, and saliency-based low-rank adapters into an integrated pipeline. First, SLIM applies SLIM-Quant , a novel scheme designed to minimize quantization error, laying the foundation for subsequent pruning using methods such as Wanda (Sun et al., 2023). Finally, low-rank adapters are introduced to reduce the impact of compression errors from both quantization and pruning, ensuring minimal accuracy loss. The overall process is illustrated in Figure 1, providing a visual summary of how these components interact to achieve effective model compression. In the following subsections, we dive into the details of each step, highlighting the innovations and contributions of SLIM-Quant , the pruning strategy, and the saliency-based low-rank adapters.

### 3.1. SLIM-Quant Quantization Method

SLIM adopts symmetric weight quantization due to its low dequantization and memory overhead and ease of implementation. Denoting the quantized matrices by $Q$ superscript, Equation 2 shows the symmetric quantization formula for $q$-bit quantization, where $\alpha$ is the quantization scaling parameter and $clip(.)$ operator clips the input to values between $[-1, 1]$.

$$\mathcal{W}^Q = round(clip(\frac{\mathcal{W}}{\alpha}))2^{q-1} \quad (2)$$

The objective of quantization is to reduce the weight reconstruction error shown in Equation 3, where the $*$ superscript

shows the optimal value. But the objective function in Equation 3 is not convex, and to our best knowledge, does not have a closed form solution.

$$\alpha^* = \arg \min_{\alpha} ||\mathcal{W}^Q - \mathcal{W}||^2$$
$$= \arg \min_{\alpha} ||round(clip(\frac{\mathcal{W}}{\alpha}))2^{q-1} - \mathcal{W}||^2 \quad (3)$$

To solve the mean squared error (MSE) problem in Equation 3, we propose a probabilistic reformulation as shown in Equation 4, where $Q(.)$ and $Q^{-1}(.)$ are the quantization and dequantization functions respectively, and $f(.)$ is the probability distribution function (PDF) of the weight elements.

$$\alpha^* = \arg \min_{\alpha} E_Q = \arg \min_{\alpha} ||\mathcal{W}^Q - \mathcal{W}||^2$$
$$= \arg \min_{\alpha} \int_{-\infty}^{\infty} f(x)|Q^{-1}(Q(x)) - x|^2 dx \quad (4)$$

By incorporating the quantization formula from Equation 2 into Equation 4, we can simplify the integration into the sum of two terms based on the absolute value of the data: the quantization error for absolute values less than $\alpha$ (Equation 5) and the clipping error for absolute values larger than $\alpha$ (Equation 6). Here, $f_{abs}(.)$ represents the probability density function (PDF) of the absolute value of the weights. Equation 7 presents the simplified version of Equation 4.

$$E_{quant}(\alpha)$$
$$= \int_0^{\alpha} f_{abs}(x)|\alpha \times round(\frac{x}{\alpha}) \times 2^{1-q} - x|^2 dx \quad (5)$$

$$E_{clip}(\alpha) = \int_{\alpha}^{\infty} f_{abs}(x)|\alpha - x|^2 dx \quad (6)$$

$$\alpha^* = \arg \min_{\alpha} E_Q(\alpha)$$
$$= \arg \min_{\alpha} E_{quant}(\alpha) + E_{clip}(\alpha) \quad (7)$$

Equation 7 can be solved theoretically by differentiating the objective function with respect to $\alpha$, provided the probability density function (PDF) of the weight distribution is known. However, the weight distribution of neural networks rarely conform to standard PDFs. To verify this, we tested various candidate distributions, including Gaussian, Laplace, Pareto, q-Gaussian, and Weibull, as they are commonly used in modeling natural data. Unfortunately, none

---

**Algorithm 1** SLIM-Quant Algorithm

1: **Input:** Weight Magnitude PDF: $f_{abs}$, High Resolution Step Size: $\eta_{high}$, Low Resolution Step Size: $\eta_{low}$ Weight Matrix: $\mathcal{W}$, Quantization Bitwidth: $q$
2: **Output:** $\mathcal{W}_{quant}$
3: **function** EstimateError($\alpha$)
4:    $E_{quant}(\alpha) = \int_0^{\alpha} f_{abs}(x)|\alpha \times round(\frac{x}{\alpha}) \times 2^{1-q} - x|^2 dx$
5:    $E_{clip}(\alpha) = \int_{\alpha}^{\infty} f_{abs}(x)|\alpha - x|^2 dx$
6:    **return** $E_{quant} + E_{clip}$
7: **end function**
8: $E$ = EmptyDictionary()
9: **for** $\alpha$ in range(0, $M$, $\eta_{low}$) **do**
10:    $E(\alpha)$ = EstimateError($\alpha$)
11: **end for**
12: $\alpha_{low} = \arg \min_{\alpha} E(\alpha)$
13: **for** $\alpha$ in range($\alpha_{low} - \eta_{low}, \alpha_{low} + \eta_{low}, \eta_{high}$) **do**
14:    $E(\alpha)$ = EstimateError($\alpha$)
15: **end for**
16: $\alpha^* = \arg \min_{\alpha} E(\alpha)$
17: $\mathcal{W}_{quant} = round(clip(\frac{W}{\alpha^*})) \times 2^{q-1}$

---

of these matched the observed weight distributions accurately. This discrepancy underscores the need for a more adaptable method, motivating the data-driven approach we adopt in SLIM-Quant .

To address the absence of a closed-form weight PDF, we employ numerical integration on the weight histogram to solve Equation 7. To enhance efficiency, we adopt a multi-grid strategy: starting with 10 uniform samples in the range $(0, \max(W))$, the grid is iteratively refined around the region of minimum error. This iterative process converges to the optimal $\alpha$ with minimal computational overhead. The full procedure is detailed in Algorithm 1.

**Activation-aware SLIM-Quant.** Recent work has shown that the quantization of a subset of the weight channels has a higher impact on output error of the model (Xiao et al., 2023; Lin et al., 2024). We extend SLIM-Quant by incorporating an output error minimization approach inspired by AWQ (Lin et al., 2024). Similar to AWQ, our revised algorithm applies a scaling strategy to activations, reducing the quantization error of salient weight channels. Specifically, we scale up the weights associated with the most significant channels and correspondingly scale down the related input activations. This approach maintains computational equivalence while effectively lowering the quantization-induced output error. In particular, scaling approximately 1% of the channels does not alter the overall quantization parameters but significantly reduces errors in the critical channels.

However, our approach diverges from AWQ by introducing a novel saliency metric that jointly considers both activations and weights. We define the saliency of each channel as the product of the normalized average magnitudes of inputs and weights, expressed as $|diag(\mathbf{x}) \times \mathcal{W}|$, where $\mathbf{x}$ and $\mathcal{W}$ denote the average magnitude of activations and weights, respectively, and $|.|$ denotes the element-wise absolute value operator. Channels with the highest saliency are scaled by a factor of $s > 1$, while their corresponding activations

are scaled by $\frac{1}{s}$. Although this method introduces modest computational overhead, that is attributed to on-the-fly adjustments of roughly 1% of activations and resulting irregular memory access patterns, it yields measurable accuracy improvements. These results underscore a clear trade-off between computational complexity and model performance, highlighting the relative strength of SLIM-Quant$^O$ (SLIM-Quant with output error minimization) over SLIM-Quant$^W$ (SLIM-Quant with weight error minimization). Please note that SLIM-Quant without the superscript $^W$ denotes the weight error minimization version of SLIM-Quant .

### 3.2. SLIM-LoRA Low-rank Adapters

After quantizing the model using SLIM-Quant , we sparsify it using an off-the-shelf one-shot pruning method such as Wanda. The combined effects of quantization and pruning of a weight matrix can be modeled as additive noise, such that $\mathcal{W}^C = \mathcal{W} + E_Q + E_S$, where $E_Q = \mathcal{W} - \mathcal{W}^Q$ and $E_S = \mathcal{W}^C - \mathcal{W}^Q$ are the quantization and sparsity errors respectively. To mitigate these errors, we introduce low-rank adapters that adjust the compressed weights such that $\mathcal{W} \approx \mathcal{W}^C + \mathcal{L}\mathcal{R}$, where $\mathcal{L} \in \mathbb{R}^{d_{in} \times r}$ and $\mathcal{R} \in \mathbb{R}^{r \times d_{out}}$ are the low-rank adapters and $r$ is the adapter rank.

A straightforward approach minimizes the total error norm between $\mathcal{W}$ and $\mathcal{W}^C$, focusing solely on reducing the error magnitude while ignoring the saliency of individual elements in the weight matrix. We call this method **Naive-LoRA** *as it overlooks the importance of individual elements in the weight matrix*. However, this method is suboptimal and can be substantially improved.

To address the limitations of Naive-LoRA, we propose a novel low-rank approximation formulation that integrates weight saliency and uses a carefully designed saliency function to determine optimal adapters. The saliency function ($F$) in our formulation needs to satisfy two key properties. First, it needs to be invertible, enabling the retrieval of low-rank adapters from their saliency. Second, it must be additive, meaning $\forall A, B : F(A + B) = F(A) + F(B)$. The additive property is crucial for isolating the saliency of low-rank adapters from the compressed matrix and distinguishing the saliency of the error from that of the original weights. These properties ensure that the saliency function can effectively isolate and optimize the contribution of low-rank adapters, forming the foundation of our proposed formulation.

Assuming that there exists an additive invertible saliency function $F : \mathbb{R}^{d_{in} \times d_{out}} \to \mathbb{R}^{d_{in} \times d_{out}}$, we need to solve Equation 8 to find the optimal adapters. Using the additive property of the saliency function $F(.)$, we can simplify Equation 8 to Equation 9.

$$\mathcal{L}, \mathcal{R} = \arg\max_{\mathcal{L}, \mathcal{R}} ||F(\mathcal{W}^C + \mathcal{L}\mathcal{R})||^2$$
$$= \arg\min_{\mathcal{L}, \mathcal{R}} ||F(\mathcal{W} - (\mathcal{W}^C + \mathcal{L}\mathcal{R}))||^2 \quad (8)$$

$$\mathcal{L}, \mathcal{R} = \arg\min_{\mathcal{L}, \mathcal{R}} ||F(\mathcal{W} - \mathcal{W}^C) - F(\mathcal{L}\mathcal{R})||^2$$
$$= \arg\min_{\mathcal{L}, \mathcal{R}} ||F(-(E_Q + E_S)) - F(\mathcal{L}\mathcal{R})||^2 \quad (9)$$

Now, we can find $F(\mathcal{L}\mathcal{R})$ by computing the SVD of $F(-(E_Q + E_S))$, and using the invertibility property of $F$, we can obtain the exact value of $\mathcal{L}$ and $\mathcal{R}$.

The saliency function used in SLIM must satisfy three essential criteria—invertibility, additivity, and the effective utilization of input and weight statistics—to optimize weight importance during compression. Recent works such as Wanda, AWQ, LLM.int8(), and L$^2$QER suggest that the product of the magnitude of the weights and activations is a useful metric for identifying important weights during pruning and quantization. Motivated by this observation, we propose a saliency function formulation for $F$ that meets these criteria and leverages weight-activation interactions for effective compression.

To incorporate input statistics into the saliency function, we define $F(\mathcal{W}) \triangleq diag(\mathbf{x})\mathcal{W}$, where $\mathbf{x} \in \mathbb{R}^{d_{in}}$ represents the average absolute value of inputs from a calibration set. This formulation ensures that the saliency function effectively weights the matrix elements based on their significance during compression, facilitating a more accurate approximation. By replacing $F(\mathcal{W})$ in Equation 9, the optimization problem transforms into a computationally efficient solution using singular value decomposition, followed by an inverse saliency transformation to derive the left low-rank adapter (Equation 11).

$$\mathcal{L}, \mathcal{R} = \arg\min_{\mathcal{L}, \mathcal{R}} || - diag(\mathbf{x})(E_Q + E_S) - diag(\mathbf{x})\mathcal{L}\mathcal{R}||^2 \quad (10)$$

$$diag(\mathbf{x})\mathcal{L}, \mathcal{R} = -SVD(diag(\mathbf{x})(E_Q + E_S)) \quad (11)$$

We refer to this method of computing saliency-based low-rank adapters as **SLIM-LoRA**, a practical and efficient approach tailored for addressing compression errors in large language models. To ensure numerical stability and guarantee the invertibility of the saliency function, an identity matrix with small values can be added to $diag(\mathbf{x})$. This adjustment is equivalent to uniformly shifting all elements of $\mathbf{x}$ and ensures that the saliency function remains robust even when $\mathbf{x}$ contains near-zero elements. Algorithm 2 provides

**Algorithm 2** SLɪM-LoRA Saliency-based Low-rank Adapter Computation

1: **Input:** Original Weight: $\mathcal{W}$, Compressed Weight: $\mathcal{W}^C$ Calibration Input: $\mathcal{X}$
2: **Output:** $\mathcal{L}, \mathcal{R}$: Saliency-based Low-rank Adapters
3: $E_C = E_Q + E_S = \mathcal{W}^C - \mathcal{W}$       // Compute Error
4: $\tilde{\mathbf{x}} = mean(\mathcal{X})$       // Average over all the samples
5: $\mathbf{x} = \tilde{\mathbf{x}} + min(|\tilde{\mathbf{x}}|)$       // Shift values to avoid zeros in $\mathbf{x}$
6: $\mathcal{S}_C = diag(\mathbf{x})E_C$       // Compute error saliency
7: $\tilde{\mathcal{L}}, \tilde{\mathcal{R}} = SVD(\mathcal{S}_C)$       // Low-rank approximation
8: $\mathcal{L} = diag(1/\mathbf{x})\tilde{\mathcal{L}}$       // Converting saliency to weight
9: $\mathcal{R} = \tilde{\mathcal{R}}$

a comprehensive overview of the steps involved in computing saliency-based low-rank adapters using SLɪM-LoRA, ensuring reproducibility and clarity.

### 3.3. Low-rank Adapter Quantization

While pruning and quantizing the weights significantly reduce the model's computation and memory requirements ($\sim 8\times$ memory footprint reduction), incorporating full-precision low-rank adapters reintroduces overhead, partially offsetting these gains. To address this, we applied 4-bit quantization to compress the adapters. This step ensures that the compression efficiency achieved through weight pruning and quantization is preserved, while maintaining the performance benefits of the low-rank adapters.

Quantizing low-rank adapters poses unique challenges due to the long-tailed distribution of their elements, which limits the effectiveness of advanced non-group quantization methods, such as SLɪM-Quant. To address this, we adopt an AbsMax group quantization scheme for the adapters, where groups of 128 elements share the same quantization parameter. By grouping elements, this method effectively captures the distribution's variability while minimizing quantization error, striking a balance between accuracy and compression. This approach not only reduces the adapter overhead by $4\times$ but ensures that their contribution to overall model compression and performance is retained; as demonstrated in our experimental evaluation.

### 3.4. Optional Post-compression Fine-tuning

Fine-tuning large language models post-compression has many challenges because the high parameter count and memory demands of traditional methods make them computationally prohibitive. For example, using a simple optimizer such as ADAMW leads to $4\times$ additional memory overhead to store gradient and optimizer states, rendering these approaches impractical for compressed models. Thus, parameter-efficient fine-tuning is essential for preserving the benefits of compression while avoiding excessive computational and memory costs. This necessity is further highlighted by the results in Section 4, which illustrate the overheads of traditional fine-tuning and the advantages of

parameter-efficient alternatives.

To overcome the challenges of fine-tuning compressed models, SLɪM employs parameter-efficient low-rank adapters as the only tunable components during the fine-tuning phase. During this optional phase, SLɪM freezes the sparse and quantized weights, enabling focused fine-tuning solely on the adapters. If the adapters are quantized, SLɪM uses a straight-through estimator (STE) for quantization-aware fine-tuning and reduces its overheads with custom quantization and dequantization kernels implemented in Triton. This parameter-efficient fine-tuning method allows rapid accuracy improvements for the compressed model, requiring only a short phase over thousands of tokens. By limiting the fine-tuning process to a small subset of parameters, SLɪM significantly reduces computational requirements while ensuring the model can adapt effectively to new data or tasks. This approach maintains the benefits of compression while enabling efficient adaptation, as demonstrated by the significant improvements achieved during fine-tuning.

## 4. Experimental Results

**Models, Datasets, and Evaluation.**[3] We evaluate SLɪM on the OPT (Zhang et al., 2022) and LLaMA-2 (Touvron et al., 2023) model families, both of which serve as standard baselines in model compression studies (Ma et al., 2024; Frantar & Alistarh, 2023; Sun et al., 2023). Model accuracy is assessed on a range of zero-shot downstream tasks, including MMLU (Hendrycks et al., 2020), Piqa (Bisk et al., 2020), Arc-Easy, Arc-Challenge (Clark et al., 2018), WinoGrande (Sakaguchi et al., 2021), and OpenBookQA (Mihaylov et al., 2018). For zero-shot evaluations, we utilize the Language Model Evaluation Harness (Gao et al., 2024) framework. In line with prior work (Sun et al., 2023; Frantar & Alistarh, 2023; Ma et al., 2024), we also report the perplexity of the models on a language modeling task on the WikiText2 (Merity et al., 2016) dataset, provided in Appendix G.

**Baselines.** We compare SLɪM against state-of-the-art one-shot pruning methods, including Wanda (Sun et al., 2023), SparseGPT (Frantar & Alistarh, 2023), and Magnitude Pruning (Han et al., 2015), as well as one-shot quantization techniques like OPTQ (Frantar et al., 2022), OmniQuant (Shao et al., 2023), AffineQuant (Ma et al., 2024), $L^2$QER (Zhang et al., 2024a), and AbsMax. Additionally, we extend Joint Sparsification and Quantization (JSQ) (Guo et al., 2024) to support 4-bit weight quantization and include it in our experiments. To ensure fairness, we use the optimal hyperparameters reported for each method, or the default hyperparameters if not explicitly reported. For a thorough **description of the notations** used to show the dif-

---
[3]All the experiments in the paper were run at the University of Toronto

ferent variants of SLIM, please see Table 4 in Appendix A. For more details about the hyperparameters used in different experiments, please see Appendix T.

$L^2QER$ (Zhang et al., 2024a) and OATS (Zhang & Papyan, 2024) are the two independent and concurrent compression methods utilizing zero-shot low-rank adapters to enhance model accuracy. Our approach, SLIM, significantly diverges from them in several key aspects. First, we employ saliency-based low-rank adapters to mitigate compression loss in *quantized and sparse* models, whereas $L^2QER$ is tailored exclusively for quantization, resulting in reduced accuracy when combined with sparsity, as demonstrated in the subsequent subsections, and OATS is designed for unstructured sparsity only without quantization, which does not have acceleration support on NVIDIA GPUs. Second, we introduce SLIM-Quant , which lowers the overhead and complexity of group quantization compared to methods like $L^2QER$. Finally, SLIM compresses and fine-tunes low-rank adapters efficiently to minimize overhead. In contrast, $L^2QER$ and OATS rely on full-precision low-rank adapters, which incur additional overhead and do not benefit from the parameter-efficient fine-tuning proposed in our work.

**Accuracy Results.** We evaluate the accuracy of SLIM and other state-of-the-art pruning and quantization methods across 2:4 and unstructured sparsity benchmarks, highlighting SLIM's superiority in Table 1. SparseGPT and Group OPTQ, designed to work together, achieve competitive performance. For other advanced quantization methods, we pruned models using Wanda and quantized the sparse checkpoints with Group AbsMax, AWQ, OmniQuant, and AffineQuant, reporting the best results (detailed in Appendix H). In particular, methods such as OmniQuant and AffineQuant struggle to quantize OPT-350M, often resulting in NaN values. Moreover, AWQ, OmniQuant, AffineQuant, and $L^2QER$ encounter out-of-memory (OOM) errors when compressing models on a single A100-40GB GPU. While JSQ performs well for the LLaMA-2 family, its difficulty compressing the OPT family limits its broader applicability.

The progression from Naive-LoRA to SLIM-LoRA and SLIM-LoRA$^Q$ demonstrates the benefits of incorporating weight saliency into low-rank adapters and applying quantization for reducing overhead. While Naive-LoRA improves model accuracy across different sizes, SLIM-LoRA achieves additional gains by effectively leveraging the saliency of the weights in the adapter design. Extending this, SLIM-LoRA$^Q$ applies quantization to the low-rank adapters, further minimizing overhead with minimal impact on accuracy, adding negligible improvements or degradation to the accuracy of the model.

Table 2 demonstrates how lightweight fine-tuning (FT) improves the accuracy of both SLIM-LoRA and Naive-LoRA, with SLIM-LoRA exhibiting greater gains due to

its saliency-aware design. Further details on the fine-tuning process and its overhead are provided in Appendix K, illustrating its practicality for enhancing compressed model performance.

**SLIM with MaskLLM.** MaskLLM is the state-of-the-art pruning method designed for 2:4 sparsity. It keeps the original weights in the model intact, while finding the optimal 2:4 masks for the model through a mask training phase. As a result, it can be combined with SLIM to boost the accuracy of the models even further. Table 3 summarizes the average accuracy results of MaskLLM on six zero-shot downstream tasks and its perplexity on WikiText2 dataset.

**Large Compressed vs. Small Dense Models.** This section compares large compressed models with dense models of equivalent parameter size, offering guidelines for configuration selection under hardware constraints. We focus on 2:4 sparsity due to its hardware acceleration support and evaluate the OPT model family, which spans a wide range of sizes for comprehensive analysis.

We analyze model performance by plotting average accuracy against parameter size, calculated as detailed in Appendix L. This visualization enables a direct performance comparison between models with an equal number of bits.

Figure 2 presents the accuracy results of the OPT model family across different compression methods. The x-axis represents the model parameter size in gigabytes, while the y-axis denotes accuracy (higher is better). The results demonstrate that SLIM-LoRA$^Q$, both with and without fine-tuning, consistently outperforms dense models and other compression techniques at the same parameter size. Notably, compressed models achieve higher accuracy than dense models of equivalent size, highlighting the effectiveness of the proposed method. This trend underscores the advantage of SLIM-LoRA$^Q$ in maximizing model efficiency under strict hardware constraints.

**Speedup.** Leveraging sparsity and quantization enhances GPU resource utilization, enabling faster model inference. Following Wanda's experimental setup, we evaluate the speedup achieved across different model layers and sizes. Similar to Wanda, AWQ, and QuaRot (Ashkboos et al., 2024), we focus on consumer-grade GPUs and conduct our experiments on NVIDIA RTX 3060 GPUs. Speedup results for NVIDIA A100 GPUs are provided in Appendix J.

SLIM achieves notable speedups through optimized sparse and quantized matrix multiplication, utilizing Sparse Marlin (Frantar et al., 2024) integrated with vLLM (Kwon et al., 2023). For inference, we adopt small batch sizes during decoding, as recommended by prior works (Xia et al., 2023; Zheng et al., 2022). Dense Quantized Marlin or PyTorch kernels handle the low-rank adapters based on their quantization status. Figure 3 highlights the speedup achieved

*Table 1.* Average zero-shot accuracy of LLaMA-2 and OPT models with **50% sparsity and 4-bit weight quantization**. *Best Method** indicates the best quantization method out of Group AbsMax, AWQ, OmniQuant, and AffineQuant. ↑ indicates better performance.

| Pruning/LoRA Method | Weight Quantization | OPT | | | | | | LLaMA-2 | |
|---|---|---|---|---|---|---|---|---|---|
| | | 125M | 350M | 1.3B | 2.7B | 6.7B | 13B | 7B | 13B |
| Dense | - | 35.9 | 37.1 | 43.4 | 45.5 | 48.3 | 48.7 | 56.6 | 60.8 |
| **2:4 Sparsity** | | | | | | | | | |
| Magnitude | Group AbsMax | 32.19 | 31.94 | 33.82 | 33.43 | 34.81 | 34.68 | 44.64 | 44.18 |
| SparseGPT | Group OPTQ | 33.70 | 33.38 | 38.75 | 40.15 | 44.32 | 45.64 | 45.49 | 51.05 |
| Wanda | Best Method* | 33.39 | 32.79 | 38.43 | 40.00 | 43.41 | 44.07 | 44.86 | 48.94 |
| JSQ | JSQ | 32.30 | 31.84 | 35.23 | 32.89 | 38.06 | 37.24 | 44.80 | 50.20 |
| $L^2$QER | Group AbsMax | 33.34 | 31.68 | 36.68 | 38.11 | 41.37 | OOM | 43.77 | OOM |
| Naive-LoRA | SLIM-Quant$^W$ | 34.28 | 33.38 | 38.36 | 41.21 | 44.91 | 45.25 | 48.45 | 51.94 |
| SLIM-LoRA | SLIM-Quant$^W$ | **34.62** | **34.36** | **40.61** | **42.73** | 45.99 | 46.09 | **51.15** | **54.94** |
| SLIM-LoRA$^Q$ | SLIM-Quant$^W$ | 34.43 | 34.30 | 40.11 | 42.37 | **46.33** | **46.24** | 51.02 | 53.55 |
| **50% Unstructured** | | | | | | | | | |
| Magnitude | Group AbsMax | 33.34 | 33.51 | 32.12 | 39.90 | 36.44 | 32.33 | 47.03 | 51.04 |
| SparseGPT | OPTQ | 35.10 | 35.13 | 38.72 | 43.43 | 46.97 | 47.38 | 51.09 | 55.94 |
| Wanda | Best Method* | 35.11 | 33.89 | 41.02 | 42.89 | 46.52 | 46.84 | 53.62 | 56.76 |
| JSQ | JSQ | 32.14 | 30.34 | 38.86 | 35.48 | 42.75 | 30.73 | 52.25 | 57.00 |
| $L^2$QER | Group AbsMax | 34.45 | 34.45 | 38.38 | 41.28 | 45.08 | OOM | 50.60 | OOM |
| Naive-LoRA | SLIM-Quant$^W$ | 34.77 | 34.23 | 40.40 | 43.37 | 46.64 | 47.30 | 51.52 | 55.33 |
| SLIM-LoRA | SLIM-Quant$^W$ | 35.20 | **35.32** | **41.85** | 43.48 | 47.08 | **47.96** | **54.26** | **57.85** |
| SLIM-LoRA$^Q$ | SLIM-Quant$^W$ | **35.35** | 35.13 | 41.74 | **43.63** | **47.16** | 47.86 | 54.18 | 57.33 |

*Table 2.* Effects of fine-tuning on the average zero-shot accuracy of LLaMA-2 models with. ↑ indicates better performance.

| Pruning/LoRA Method | Weight Quantization | LLaMA-2 | |
|---|---|---|---|
| | | 7B | 13B |
| Dense | - | 56.6 | 60.8 |
| **50% 2:4** | | | |
| Naive-LoRA + FT | SLIM-Quant$^W$ | 50.89 | 55.70 |
| SLIM-LoRA + FT | SLIM-Quant$^W$ | **52.12** | **56.60** |
| SLIM-LoRA$^Q$ + FT | SLIM-Quant$^W$ | 48.31 | 56.50 |
| **50% Unstructured** | | | |
| Naive-LoRA + FT | SLIM-Quant$^W$ | 52.90 | 57.08 |
| SLIM-LoRA + FT | SLIM-Quant$^W$ | **54.69** | **57.96** |
| SLIM-LoRA$^Q$ + FT | SLIM-Quant$^W$ | 53.57 | 57.78 |

*Table 3.* Accuracy (Acc) and perplexity (PPL) of MaskLLM combined with SLIM on LLaMA-2-7B.

| Pruning/LoRA Method | Weight Quantization | LLaMA-2-7B | |
|---|---|---|---|
| | | Acc | PPL |
| Dense | - | 56.6 | 5.47 |
| MaskLLM | - | 49.7 | 7.3 |
| Naive-LoRA | - | 52.3 | 6.6 |
| SLIM-LoRA | - | 52.2 | 7.0 |
| Naive-LoRA + FT | - | 52.6 | 6.6 |
| SLIM-LoRA + FT | - | 52.9 | 6.6 |
| MaskLLM | Group AbsMax | 49.2 | 7.6 |
| Naive-LoRA | SLIM-Quant | 50.5 | 7.3 |
| SLIM-LoRA | SLIM-Quant | 51.4 | 7.5 |
| Naive-LoRA + FT | SLIM-Quant | 51.2 | 6.9 |
| SLIM-LoRA + FT | SLIM-Quant | 52.1 | 6.8 |

across different LLaMA-2 layers compared to dense, unquantized models. The breakdown of the speedup, showing the contribution of the quantization and sparsity, is demonstrated using brighter and darker colors respectively. Larger matrices, such as those in feed-forward modules, consistently yield greater speedups, aligning with trends detailed in Appendix J.

**Model Memory Reduction.** We evaluate SLIM's memory reduction on A100 GPUs and our experiments show that SLIM$^Q$ achieves $0.23\times$ and $0.23\times$ memory reduction on LLaMA-2-7B and LLaMa-2-13B respectively. The reductions for SLIM are $0.33\times$ and $0.34\times$ respectively.

**Additional Experiments.** Due to the page limit, we provide additional experiments for a comprehensive evaluation in the appendix.

An evaluation of SLIM with input quantization using FP8 is provided in Input Quantizatoin (Appendix B). The results show that input quantization has a minimal impact on the accuracy of the models using SLIM .

A comparison between weight error minimization and activation error minimization in SLIM-Quant is provided in SLIM-Quant$^W$ vs. SLIM-Quant$^O$ (Appendix C). The exper-

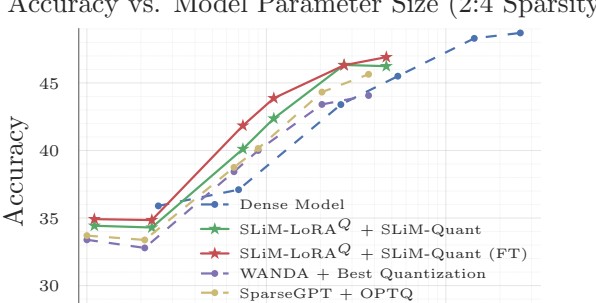

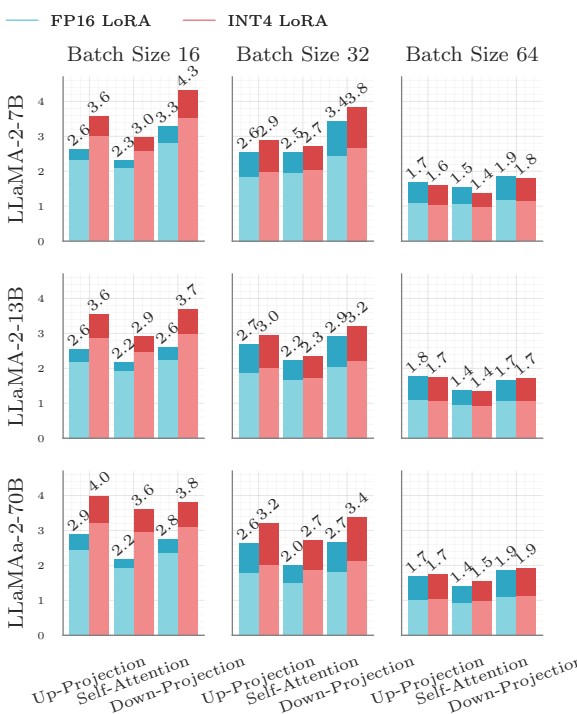

*Figure 2.* Accuracy results of the OPT family across different compression methods (↑ indicates better performance). At equal parameter size, SLiM outperforms both dense models and other compression techniques, demonstrating that model compression with SLiM yields superior performance under the same budget.

iments show that the gap between output error minimization and weight error minimization is not significant.

We evaluate SLiM on sparse-only and quantized-only models to isolate their effectiveness. Results in Additional Sparse-only Results (Appendix D) and Additional Quantization-only Results (Appendix E) demonstrate that SLiM and SLiM-Quant consistently outperform state-of-the-art compression methods.

The Language Modeling Experiments (Appendix G) evaluates SLiM across sparse and quantized, sparse-only, and quantized-only models on WikiText-2. The results align with the accuracy trends reported in the main paper, further validating the effectiveness of SLiM .

The Fine-tuning Costs (Appendix K) shows that SLiM reduces fine-tuning overhead from over 36 days for 13B parameter models to just 14 hours on a single GPU, demonstrating its practicality and efficiency.

We provide a comparison between Sparsity vs. Quantization (Appendix I) to show that combining 50% sparsity and 4-bit quantization helps achieve better compression results in comparison to solely using 2-bit quantization, while maintaining a similar compression ratio (∼8×).

Additional speedup results for SLiM on NVIDIA A100-40GB GPUs are provided in the Additional Speedup Results (Appendix J). A theoretical analysis of computation and memory reductions can be found in the Computation Reduction Analysis (Appendix M) and Memory Reduction Analysis (Appendix L), highlighting the efficiency of SLiM.

*Figure 3.* LLaMA-2 family of models speedup (×) using SLiM compared to original dense unquantized model on NVIDIA RTX-3060. ↑ shows higher speedup. The brighter color shows the contribution of quantization to the total speedup.

Compression Costs (Appendix N) details the time required to compress models of various sizes across different methods. Rank Analysis (Appendix O) explores how rank choices in low-rank adapters impact computational and memory costs, as well as model accuracy. Sparsity Analysis (Appendix D) analyzes the effects of different sparsity ratios on model compression. Lastly, Effects of Calibration Sample Count (Appendix P) evaluates the influence of calibration sample counts on the accuracy of calibration-based methods.

Finally, more details on per-task accuracy results on downstream tasks reported in different tables, please refer to our Weights & Biases report at https://bit.ly/4oAsWhr.

## 5. Conclusion

We introduced SLiM, a one-shot quantized sparse plus low-rank approximation method for large language models, optimizing both efficiency and accuracy. By combining quantization, sparsity, and saliency-based low-rank adapters, SLiM achieves substantial reductions in memory and computation while preserving competitive performance. SLiM outperforms state-of-the-art methods in accuracy.

## Impact Statement

The SLIM framework advances model compression by enabling efficient, one-shot quantization and sparsity for large language models (LLMs) while maintaining accuracy through low-rank approximation. This has the potential to make LLMs more accessible and sustainable by reducing their computational and energy requirements, thereby enabling deployment on a wider range of devices, such as smartphones and edge computing platforms, and contributing to environmental sustainability. However, the increased accessibility of compressed models raises important considerations regarding potential accuracy trade-offs in critical applications, such as healthcare or legal systems, and the ethical implications of broader AI deployment, including risks of bias propagation and misuse. To address these challenges, it is crucial to ensure that efficiency gains do not compromise model reliability and that appropriate safeguards, such as transparency and rigorous evaluation, are in place for responsible AI development.

## Acknowledgments

This work was also supported in part by NSERC Discovery Grants (RGPIN-06516, DGECR00303), the Canada Research Chairs program, Ontario Early Researcher award, the Canada Research Chairs program, the Ontario Early Researcher Award, and the Digital Research Alliance of Canada (www.alliancecan.ca). We extend our gratitude towards Ray Hung, Behrooz Zarebavani, Joan Puigcerver, James Laudon, Suvinay Subramanian, and Cliff Young for reviewing the paper and providing insightful feedback. We also thank the extended team at Google DeepMind who enabled and supported this research direction.

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

# Appendix

## A. Notations

Table 4 details the key notations, particularly for Section 4.

*Table 4.* Group quantization slow-down on different LLaMA-2 and LLaMA 3.1 models.

| Term | Description |
|---|---|
| Naive-LoRA | A one-shot low-rank adapter that minimizes the norm of the difference between the original and the compressed weights. |
| SLɪM-LoRA | A saliency-based one-shot low-rank adapter that minimizes the saliency of the difference between the original and the compressed weights. |
| $Q$ (Superscript) | $^Q$ indicates that the compression method quantizes the low-rank adapters as well. |
| + FT | + FT shows a short fine-tuning phase on 300,000 tokens from the C4 dataset. |

## B. Input Quantization

We evaluate SLɪM with 8-bit input quantization to assess its impact on accuracy. We use AbsMax uniform quantization with a single parameter per input tensor and apply FP8 format (Micikevicius et al., 2022) for weight quantization. The choice between E4M3 and E5M2 depends on the tensor's maximum value; if it exceeds E4M3's range, we switch to E5M2 for greater expressivity. Next, we examine how input quantization affects model accuracy.

Table 5 presents accuracy results for different SLɪM variants with input quantization. A comparison with Table 9, which reports accuracy without input quantization, reveals minimal accuracy loss, demonstrating SLɪM's robustness. For further validation, we extend these experiments to language modeling tasks (Appendix G).

*Table 5.* Average zero-shot accuracy of LLaMA-2 and OPT models with **4-bit weight quantization and 8-bit input quantization** with 50% weight sparsity. ↑ indicates better performance.

| Pruning/LoRA Method | Weight Quantization | OPT | | | | | | LLaMA-2 | |
|---|---|---|---|---|---|---|---|---|---|
| | | 125M | 350M | 1.3B | 2.7B | 6.7B | 13B | 7B | 13B |
| Dense | - | 35.9 | 37.1 | 43.4 | 45.5 | 48.3 | 48.7 | 56.6 | 60.8 |
| **50% 2:4** | | | | | | | | | |
| SLɪM-LoRA | SLɪM-Quant$^W$ | 34.85 | 34.27 | 40.29 | 42.58 | 45.78 | 46.21 | 50.99 | 54.66 |
| SLɪM-LoRA + FT | SLɪM-Quant$^W$ | 35.28 | 34.33 | 41.14 | 43.29 | 46.44 | 47.33 | 51.77 | 56.28 |
| SLɪM-LoRA$^Q$ | SLɪM-Quant$^W$ | 34.30 | 33.85 | 39.92 | 41.99 | 46.08 | 45.94 | 50.70 | 53.56 |
| SLɪM-LoRA$^Q$ + FT | SLɪM-Quant$^W$ | 34.92 | 34.80 | 41.66 | 43.69 | 46.03 | 46.87 | 50.26 | 56.28 |
| **50% Unstructured** | | | | | | | | | |
| SLɪM-LoRA | SLɪM-Quant$^W$ | 35.12 | 34.86 | 41.94 | 43.53 | 47.27 | 47.70 | 54.28 | 57.82 |
| SLɪM-LoRA + FT | SLɪM-Quant$^W$ | 35.18 | 35.30 | 42.37 | 44.02 | 47.01 | 48.52 | 54.43 | 57.70 |
| SLɪM-LoRA$^Q$ | SLɪM-Quant$^W$ | 35.26 | 34.67 | 41.48 | 43.46 | 47.25 | 47.76 | 53.91 | 57.16 |
| SLɪM-LoRA$^Q$ + FT | SLɪM-Quant$^W$ | 35.52 | 35.31 | 42.66 | 44.50 | 47.08 | 48.53 | 53.23 | 57.55 |

## C. SLɪM-Quant$^W$ vs. SLɪM-Quant$^O$

Table 6 compares the average accuracy of different models when using SLɪM-Quant with weight error minimization (SLɪM-Quant$^W$) and activation-aware output error minimization (SLɪM-Quant$^O$). SLɪM-Quant$^O$ outperforms SLɪM-Quant$^W$ by a small gap, while adding computational overhead with irregular memory access patterns at inference time.

*Table 6.* Average zero-shot accuracy of LLaMA-2 and OPT models with **50% sparsity and 4-bit weight quantization** for SLıM-Quant$^W$ and SLıM-Quant$^O$.

| Pruning/LoRA | Weight | OPT | | | | | | LLaMA-2 | |
| Method | Quantization | 125M | 350M | 1.3B | 2.7B | 6.7B | 13B | 7B | 13B |
|---|---|---|---|---|---|---|---|---|---|
| Dense | - | 35.9 | 37.1 | 43.4 | 45.5 | 48.3 | 48.7 | 56.6 | 60.8 |
| **2:4 Sparsity** | | | | | | | | | |
| SLıM-LoRA | SLıM-Quant$^W$ | 34.62 | **34.36** | **40.61** | **42.73** | **45.99** | 46.09 | 51.15 | 54.94 |
| SLıM-LoRA | SLıM-Quant$^O$ | **34.63** | **34.36** | 40.29 | 42.45 | 45.71 | **46.24** | **51.22** | **55.05** |
| **50% Unstructured** | | | | | | | | | |
| SLıM-LoRA | SLıM-Quant$^W$ | **35.20** | **35.32** | **41.85** | **43.48** | 47.08 | **47.96** | 54.26 | 57.85 |
| SLıM-LoRA | SLıM-Quant$^O$ | **35.20** | 34.78 | 41.29 | 43.31 | **47.09** | 47.86 | **54.46** | **57.97** |

# D. Additional Sparse-only Results

To evaluate the isolated impact of sparsity on model accuracy, we disable quantization and benchmark Magnitude Pruning, SparseGPT, and Wanda, alongside low-rank approximations like Wanda-SVD and SLıM . Our experiments assess both 50% unstructured sparsity and 2:4 structured sparsity patterns.

Table 7 shows the accuracy results for sparse models. Magnitude Pruning performs the worst, while Wanda and SparseGPT achieve comparable results, with larger accuracy gaps for semi-structured sparsity. Low-rank adapters improve accuracy, with SLıM leveraging saliency-based approximation for superior performance. A brief fine-tuning phase further boosts the accuracy of low-rank approximations.

*Table 7.* Average zero-shot accuracy of LLaMA-2 and OPT models with pruning. The quantization is disabled in this experiment. ↑ indicates better performance.

| Pruning/LoRA | OPT | | | | | | LLaMA-2 | |
| Method | 125M | 350M | 1.3B | 2.7B | 6.7B | 13B | 7B | 13B |
|---|---|---|---|---|---|---|---|---|
| Dense | 35.9 | 37.1 | 43.4 | 45.5 | 48.3 | 48.7 | 56.6 | 60.8 |
| **2:4 Sparsity** | | | | | | | | |
| Magnitude | 32.6 | 31.8 | 35.4 | 33.9 | 36.4 | 30.7 | 31.2 | 32.0 |
| SparseGPT | 33.8 | 33.2 | 37.7 | 41.3 | 45.2 | 45.6 | 47.3 | 52.3 |
| Wanda | 34.0 | 32.5 | 38.3 | 40.5 | 43.2 | 44.1 | 46.1 | 49.7 |
| SLıM-Naive | 34.1 | 34.1 | 40.4 | 42.8 | 46.0 | 45.9 | 51.6 | 55.8 |
| SLıM-Naive + FT | 34.8 | 34.5 | 41.3 | 43.4 | **46.5** | 47.2 | **52.4** | **56.9** |
| SLıM-LoRA | 34.5 | 32.9 | 40.7 | 43.1 | 46.4 | 46.3 | 51.4 | 56.1 |
| SLıM-LoRA + FT | **35.1** | **34.9** | **41.5** | **43.8** | **46.5** | **47.3** | 51.6 | 56.4 |
| **50% Unstructured** | | | | | | | | |
| Magnitude | 33.3 | 33.7 | 34.0 | 40.6 | 35.8 | 30.9 | 32.6 | 31.9 |
| SparseGPT | 35.5 | 35.1 | 39.6 | 43.5 | 47.4 | 47.8 | 53.3 | 57.3 |
| Wanda | 35.0 | 34.5 | 41.1 | 42.9 | 46.5 | 46.8 | 52.7 | 57.2 |
| SLıM-Naive | 35.3 | 35.2 | 41.9 | 44.1 | 47.5 | 47.8 | 54.9 | 58.5 |
| SLıM-Naive + FT | 35.74 | 35.7 | 42.7 | 44.6 | **47.8** | **48.4** | 54.9 | 58.7 |
| SLıM -LoRA | 35.2 | 35.1 | 42.0 | 44.1 | 47.7 | 48.2 | **55.0** | **58.8** |
| SLıM -LoRA + FT | **35.9** | **35.7** | **42.5** | **44.7** | 47.7 | **48.4** | **55.0** | **58.8** |

# E. Additional Quantization-only Results

To evaluate the impact of SLıM-Quant and low-rank compensation in SLıM, we conduct experiments without sparsity, testing quantization schemes like Group AbsMax, OPTQ, AWQ, OmniQuant, AffineQuant, $L^2$QER, and SLıM-Quant . To enhance accuracy, we add low-rank adapters to SLıM-Quant and Group AbsMax, optimizing either error saliency (SLıM-LoRA) or reconstruction error norm (Naive-LoRA). Other quantization methods cannot incorporate low-rank adapters due to conflicting weight/activation update rules.

Table 8 presents the quantization results. Adding low-rank adapters to Group AbsMax significantly boosts model accuracy, outperforming most advanced methods. While SLıM-Quant alone is not designed for high accuracy, its integration with SLıM variants achieves results comparable to or better than Group AbsMax with low-rank adapters, highlighting the value of co-design in compression methods. Furthermore, a lightweight fine-tuning phase with SLıM-Quant delivers state-of-the-art accuracy.

*Table 8.* Average zero-shot accuracy of LLaMA-2 and OPT models with quantization. The sparsity is disabled in this experiment. ↑ indicates better performance.

| Quantization Method | Low-rank Adapter | OPT | | | | | | LLaMA-2 | |
|---|---|---|---|---|---|---|---|---|---|
| | | 125M | 350M | 1.3B | 2.7B | 6.7B | 13B | 7B | 13B |
| Dense | - | 35.9 | 37.1 | 43.4 | 45.5 | 48.3 | 48.7 | 56.6 | 60.8 |
| OPTQ | - | 35.64 | 36.46 | 42.83 | 44.20 | 47.46 | 48.24 | 53.53 | 59.80 |
| AWQ | - | 36.16 | 31.83 | 42.98 | 45.28 | 48.45 | 48.76 | 53.97 | OOM |
| OmniQuant | - | 35.46 | NaN | 42.15 | 44.71 | 46.65 | OOM | 54.33 | OOM |
| AffineQuant | - | 35.73 | NaN | 42.62 | 44.92 | 47.91 | OOM | 54.52 | OOM |
| Group AbsMax | - | 35.45 | 36.67 | 42.57 | 44.79 | 48.30 | 48.49 | 55.56 | 60.12 |
| Group AbsMax | $L^2$QER | 34.75 | 35.63 | 40.60 | 44.22 | 46.90 | OOM | 55.95 | OOM |
| Group AbsMax | SLıM-Naive | **36.30** | 36.58 | 43.07 | 45.13 | 48.26 | 48.72 | 56.23 | 60.53 |
| Group AbsMax | SLıM-LoRA | 36.18 | **36.72** | 42.89 | **45.65** | **48.45** | 48.89 | 55.99 | 60.16 |
| SLıM-Quant$^W$ | - | 31.98 | 36.46 | 36.19 | 40.08 | 45.61 | 38.27 | 31.11 | 30.51 |
| SLıM-Quant$^W$ | SLıM-Naive | 35.29 | 36.02 | 42.48 | 45.01 | 47.75 | 48.38 | 55.96 | **60.85** |
| SLıM-Quant$^W$ | SLıM-LoRA | 35.69 | 36.42 | 42.59 | 45.26 | 48.18 | 48.52 | 56.26 | 60.59 |
| SLıM-Quant$^W$ | SLıM-LoRA + FT | 35.91 | 36.61 | **43.29** | 45.58 | 48.29 | **49.04** | **56.51** | 60.65 |

# F. Additional Fine-tuning Results

To complement the results in Section 4, we provide accuracy measurements for PEFT-based fine-tuning of low-rank adapters on the OPT and LLaMA-2 model families in Table 9 while showing the accuracy results without fine-tuning for comparison. The results confirm the previously observed trend: lightweight fine-tuning enhances the accuracy of all baselines, with SLıM-LoRA achieving the most significant improvements due to its saliency-based design.

# G. Language Modeling Experiments

We evaluate all benchmarks from Section 4 and Appendix B ,D, and E on the WikiText2 language modeling task. Tables 10 and 11 show perplexity results for 4-bit quantized models with 2:4 and unstructured sparsity, respectively. Table 12 summarizes the results for 8-bit input quantization. To examine sparsity and quantization independently, Tables 13 and 14 report results for pruning-only and quantization-only models. Consistent with Section 4, SLıM achieves superior performance across all settings.

*Table 9.* Effects of fine-tuning on the average zero-shot accuracy of LLaMA-2 and OPT models with. ↑ indicates better performance.

| Pruning/LoRA Method | Weight Quantization | OPT | | | | | | LLaMA-2 | |
|---|---|---|---|---|---|---|---|---|---|
| | | 125M | 350M | 1.3B | 2.7B | 6.7B | 13B | 7B | 13B |
| Dense | - | 35.9 | 37.1 | 43.4 | 45.5 | 48.3 | 48.7 | 56.6 | 60.8 |
| **50% 2:4** | | | | | | | | | |
| Naive-LoRA | SLIM-Quant$^W$ | 34.28 | 33.38 | 38.36 | 41.21 | 44.91 | 45.25 | 48.45 | 51.94 |
| Naive-LoRA + FT | SLIM-Quant$^W$ | 34.41 | 34.70 | 39.72 | 42.88 | 46.16 | 46.76 | 50.89 | 55.70 |
| SLIM-LoRA | SLIM-Quant$^W$ | 34.62 | 34.36 | 40.61 | 42.73 | 45.99 | 46.09 | 51.15 | 54.94 |
| SLIM-LoRA + FT | SLIM-Quant$^W$ | **35.03** | 34.58 | 41.11 | 43.35 | **46.71** | **47.25** | **52.12** | **56.60** |
| SLIM-LoRA$^Q$ | SLIM-Quant$^W$ | 34.43 | 34.30 | 40.11 | 42.37 | 46.33 | 46.24 | 51.02 | 53.55 |
| SLIM-LoRA$^Q$ + FT | SLIM-Quant$^W$ | 34.92 | **34.85** | **41.84** | **43.87** | 46.31 | 46.91 | 48.31 | 56.50 |
| **50% Unstructured** | | | | | | | | | |
| Naive-LoRA | SLIM-Quant$^W$ | 34.77 | 34.23 | 40.40 | 43.37 | 46.64 | 47.30 | 51.52 | 55.33 |
| Naive-LoRA + FT | SLIM-Quant$^W$ | 35.70 | 35.47 | 41.89 | 44.16 | 47.08 | 47.78 | 52.90 | 57.08 |
| SLIM-LoRA | SLIM-Quant$^W$ | 35.20 | 35.32 | 41.85 | 43.48 | 47.08 | 47.96 | 54.26 | 57.85 |
| SLIM-LoRA + FT | SLIM-Quant$^W$ | 35.59 | **35.71** | 42.37 | **44.58** | **47.69** | 48.26 | **54.69** | **57.96** |
| SLIM-LoRA$^Q$ | SLIM-Quant$^W$ | **35.35** | 35.13 | 41.74 | 43.63 | 47.16 | 47.86 | 54.18 | 57.33 |
| SLIM-LoRA$^Q$ + FT | SLIM-Quant$^W$ | 35.65 | 35.67 | **42.74** | 44.54 | 47.48 | **48.40** | 53.57 | 57.78 |

*Table 10.* Perplexity of LLaMA-2 and OPT models with **2:4 sparsity and 4-bit weight quantization** on WikiText-2 dataset language modeling task. ↓ indicates better performance.

| Pruning/LoRA Method | Weight Quantization | OPT | | | | | | LLaMA-2 | |
|---|---|---|---|---|---|---|---|---|---|
| | | 125M | 350M | 1.3B | 2.7B | 6.7B | 13B | 7B | 13B |
| Dense | - | 27.66 | 22.00 | 14.62 | 12.47 | 10.86 | 10.13 | 5.47 | 4.89 |
| Magnitude | Group AbsMax | 5.1E2 | 4.4E2 | 1.2E3 | 1.3E3 | 3.6E2 | 4.9E2 | 86.34 | 8.98 |
| SparseGPT | Group OPTQ | 78.18 | 59.86 | 27.36 | 18.62 | 15.31 | 13.25 | 12.07 | 9.46 |
| Wanda | Group AbsMax | 1.8E2 | 1.3E2 | 32.76 | 24.48 | 17.29 | 16.86 | 14.36 | 9.38 |
| Wanda | AWQ | 9.3E1 | 8.1E5 | 29.56 | 22.91 | 16.28 | 16.72 | 12.79 | OOM |
| Wanda | OmniQuant | 9.7E1 | NaN | 33.61 | 25.89 | 19.09 | OOM | 12.77 | OOM |
| Wanda | AffineQuant | 9.7E1 | NaN | 30.32 | 1.6E3 | 16.85 | OOM | 12.21 | OOM |
| JSQ | JSQ | 3.5E3 | 1.7E4 | 1.1E2 | 6.6E2 | 36.94 | 2.3E2 | 12.68 | 8.70 |
| Naive-LoRA | Group AbsMax | 69.23 | 50.02 | 20.52 | 16.05 | 12.83 | 13.12 | 8.04 | 6.38 |
| Naive-LoRA | SLIM-Quant$^W$ | 83.08 | 58.69 | 27.06 | 20.92 | 14.29 | 13.20 | 8.19 | 7.09 |
| Naive-LoRA + FT | SLIM-Quant$^W$ | 51.82 | 38.84 | 20.59 | 16.19 | 13.13 | 12.55 | 6.96 | 6.01 |
| SLIM-LoRA | SLIM-Quant$^W$ | 57.91 | 50.09 | 19.64 | 15.65 | 12.71 | 12.13 | 7.77 | 6.80 |
| SLIM-LoRA + FT | SLIM-Quant$^W$ | 44.03 | 37.32 | 18.25 | 14.89 | 12.68 | 12.06 | **6.70** | 6.60 |
| SLIM-LoRA$^Q$ | SLIM-Quant$^W$ | 53.09 | 46.96 | 19.62 | 16.01 | 12.48 | 12.15 | 7.75 | 6.96 |
| SLIM-LoRA$^Q$ + FT | SLIM-Quant$^W$ | **42.80** | 37.39 | 18.38 | 15.40 | 12.65 | 12.35 | 7.08 | **6.36** |

*Table 11.* Perplexity of LLaMA-2 and OPT models with **unstructured sparsity and 4-bit weight quantization** on WikiText-2 dataset language modeling task. ↓ indicates better performance.

| Pruning/LoRA Method | Weight Quantization | OPT | | | | | | LLaMA-2 | |
|---|---|---|---|---|---|---|---|---|---|
| | | 125M | 350M | 1.3B | 2.7B | 6.7B | 13B | 7B | 13B |
| Dense | - | 27.66 | 22.00 | 14.62 | 12.47 | 10.86 | 10.13 | 5.47 | 4.89 |
| Magnitude | Group AbsMax | 3.2E2 | 1.1E2 | 3.2E3 | 3.6E2 | 7.2E2 | 5.4E3 | 17.18 | 6.77 |
| SparseGPT | Group OPTQ | 42.60 | 34.19 | 21.41 | 14.30 | 12.15 | 11.26 | 8.28 | 5.92 |
| Wanda | Group AbsMax | 62.64 | 39.60 | 19.93 | 15.01 | 12.31 | 12.46 | 6.80 | 5.75 |
| Wanda | AWQ | 42.49 | 3.8E5 | 18.80 | 14.67 | 12.17 | 12.34 | 7.28 | OOM |
| Wanda | OmniQuant | 43.55 | NaN | 20.58 | 15.82 | 13.29 | OOM | 7.40 | OOM |
| Wanda | AffineQuant | 43.66 | NaN | 19.40 | 14.94 | 12.39 | OOM | 7.21 | OOM |
| JSQ | JSQ | 4.2E3 | 3.3E4 | 31.78 | 1.7E2 | 19.97 | 8.9E5 | 7.17 | 6.19 |
| Naive-LoRA | Group AbsMax | 40.37 | 30.99 | 17.02 | 13.91 | 11.68 | 11.38 | 6.12 | 5.28 |
| Naive-LoRA | SLɪM-Quant$^W$ | 46.66 | 33.90 | 19.46 | 15.36 | 12.16 | 11.41 | 6.56 | 5.58 |
| Naive-LoRA + FT | SLɪM-Quant$^W$ | 38.05 | 29.27 | 17.52 | 14.39 | 12.28 | 11.84 | 6.10 | 5.28 |
| SLɪM-LoRA | SLɪM-Quant$^W$ | 39.62 | 31.51 | 16.52 | 13.65 | 11.42 | 10.82 | 6.16 | 5.36 |
| SLɪM-LoRA + FT | SLɪM-Quant$^W$ | **34.92** | **28.67** | **16.16** | **13.66** | 11.83 | 11.47 | **5.36** | **5.19** |
| SLɪM-LoRA$^Q$ | SLɪM-Quant$^W$ | 38.79 | 30.16 | 16.64 | 13.82 | 11.43 | 10.80 | 6.26 | 5.58 |
| SLɪM-LoRA$^Q$ + FT | SLɪM-Quant$^W$ | 35.17 | 28.31 | 16.46 | 13.96 | **11.42** | **10.80** | 5.94 | 5.46 |

*Table 12.* Perplexity of LLaMA-2 and OPT models with **4-bit weight quantization and 8-bit input quantization**. ↓ indicates better performance.

| Pruning/LoRA Method | Weight Quantization | OPT | | | | | | LLaMA-2 | |
|---|---|---|---|---|---|---|---|---|---|
| | | 125M | 350M | 1.3B | 2.7B | 6.7B | 13B | 7B | 13B |
| Dense | - | 27.66 | 22.00 | 14.62 | 12.47 | 10.86 | 10.13 | 5.47 | 4.89 |
| **50% 2:4** | | | | | | | | | |
| SLɪM-LoRA | SLɪM-Quant$^W$ | 48.4 | 49.6 | 16.6 | 16.2 | 12.9 | 12.3 | 7.2 | 6.5 |
| SLɪM-LoRA + FT | SLɪM-Quant$^W$ | 39.8 | 37.5 | 18.3 | 15.5 | 12.8 | 12.1 | 6.6 | 5.8 |
| SLɪM-LoRA$^Q$ | SLɪM-Quant$^W$ | 54.2 | 50.8 | 20.8 | 16.8 | 13.0 | 12.4 | 7.8 | 7.0 |
| SLɪM-LoRA$^Q$ + FT | SLɪM-Quant$^W$ | 43.4 | 39.1 | 19.3 | 16.0 | 13.1 | 12.6 | 7.1 | 5.8 |
| **50% Unstructured** | | | | | | | | | |
| SLɪM-LoRA | SLɪM-Quant$^W$ | 36.8 | 31.1 | 16.8 | 14.0 | 11.7 | 10.9 | 6.1 | 5.4 |
| SLɪM-LoRA + FT | SLɪM-Quant$^W$ | 33.8 | 28.6 | 16.5 | 14.0 | 12.0 | 11.5 | 5.9 | 5.2 |
| SLɪM-LoRA$^Q$ | SLɪM-Quant$^W$ | 39.5 | 31.3 | 17.3 | 14.2 | 11.8 | 10.9 | 6.3 | 5.6 |
| SLɪM-LoRA$^Q$ + FT | SLɪM-Quant$^W$ | 35.6 | 29.1 | 17.0 | 14.3 | 12.2 | 11.7 | 6.2 | 5.5 |

*Table 13.* Perplexity of LLaMA-2 and OPT models with pruning on WikiText-2 dataset language modeling task. The quantization is disabled in this experiment. ↓ indicates better performance.

| Pruning/LoRA | OPT | | | | | | LLaMA-2 | |
| Method | 125M | 350M | 1.3B | 2.7B | 6.7B | 13B | 7B | 13B |
|---|---|---|---|---|---|---|---|---|
| Dense | 27.66 | 22.00 | 14.62 | 12.47 | 10.86 | 10.13 | 5.47 | 4.89 |
| **2:4 Sparsity** | | | | | | | | |
| Magnitude | 341.5 | 417.1 | 427.2 | 1.2E3 | 264.1 | 4.0E4 | 9.1E4 | 2.0E5 |
| SparseGPT | 60.7 | 50.7 | 23.8 | 17.2 | 14.1 | 12.9 | 10.2 | 8.3 |
| Wanda | 81.6 | 116.0 | 27.8 | 21.4 | 16.0 | 16.4 | 12.0 | 8.5 |
| Naive-LoRA | 46.9 | 45.0 | 18.8 | 15.2 | 12.5 | 12.9 | 8.1 | 6.5 |
| Naive-LoRA + FT | 39.6 | 35.1 | **15.0** | 16.3 | 12.7 | 12.3 | 6.5 | **5.7** |
| SLıM-LoRA | 45.2 | 43.6 | 18.6 | 15.0 | **12.4** | 12.6 | 7.3 | 6.2 |
| SLıM-LoRA + FT | **37.1** | **33.7** | 17.0 | **14.2** | 12.4 | **12.1** | **6.4** | 5.8 |
| **50% Unstructured** | | | | | | | | |
| Magnitude | 193.4 | 97.8 | 1.7E3 | 265.2 | 968.7 | 2.4E4 | 9.9E4 | 1.1E5 |
| SparseGPT | 36.7 | 31.8 | 17.6 | 13.4 | 11.5 | 11.1 | 6.5 | 5.6 |
| Wanda | 39.3 | 36.4 | 18.3 | 14.3 | 12.0 | 12.3 | 6.4 | 5.4 |
| Naive-LoRA | 33.3 | 29.1 | 16.3 | 13.5 | 11.5 | 11.2 | 6.2 | 5.4 |
| Naive-LoRA + FT | 31.9 | 27.5 | 16.3 | 13.8 | 12.0 | 11.6 | 5.8 | 5.1 |
| SLıM -LoRA | 32.7 | 29.0 | 15.9 | 13.2 | **11.2** | **10.8** | 5.9 | 5.2 |
| SLıM -LoRA + FT | **31.0** | **26.8** | **15.5** | **13.1** | 11.6 | 11.0 | **5.8** | **4.7** |

*Table 14.* Perplexity of LLaMA-2 and OPT models with quantization on WikiText-2 dataset language modeling task. The sparsity is disabled in this experiment. ↑ indicates better performance.

| Quantization | Low-rank | OPT | | | | | | LLaMA-2 | |
| Method | Adapter | 125M | 350M | 1.3B | 2.7B | 6.7B | 13B | 7B | 13B |
|---|---|---|---|---|---|---|---|---|---|
| Dense | - | 27.66 | 22.00 | 14.62 | 12.47 | 10.86 | 10.13 | 5.47 | 4.89 |
| OPTQ | - | 33.0 | 24.4 | 16.0 | 13.0 | 11.3 | 10.3 | 6.1 | 4.9 |
| AWQ | - | 29.1 | 2.7E5 | 14.9 | 12.7 | 11.0 | 10.2 | 6.0 | OOM |
| OmniQuant | - | 30.2 | NaN | 15.8 | 13.3 | 11.6 | OOM | 5.7 | OOM |
| AffineQuant | - | 28.7 | NaN | 14.9 | 12.6 | 11.0 | OOM | 5.7 | OOM |
| Group AbsMax | - | 35.1 | 23.3 | 15.5 | 12.9 | 11.1 | 10.3 | 5.4 | 4.7 |
| Group AbsMax | Naive-LoRA | 30.4 | 22.9 | 15.1 | 12.7 | 11.0 | 10.2 | 5.3 | 4.7 |
| Group AbsMax | SLıM-LoRA | **29.3** | **22.8** | 15.0 | **12.7** | **10.9** | 10.2 | **5.2** | **4.7** |
| SLıM-Quant$^W$ | - | 1.4E3 | 26.0 | 1.7E3 | 33.1 | 31.0 | 6.7E2 | 1.3E5 | 7.8E4 |
| SLıM-Quant$^W$ | Naive-LoRA | 32.1 | 24.1 | 15.6 | 13.4 | 11.2 | 10.5 | 5.4 | 4.8 |
| SLıM-Quant$^W$ | SLıM-LoRA | 30.8 | 23.1 | **15.2** | 12.9 | 11.1 | 10.3 | 5.4 | 4.8 |
| SLıM-Quant$^W$ | SLıM-LoRA + FT | 30.7 | 23.5 | 15.3 | 13.3 | 11.6 | **10.0** | 5.3 | 4.7 |

## H. Additional Sparse and Quantized Results

In Section 4, we provided the accuracy results for different pruning and quantization methods. When using Wanda for pruning, we only reported the best quantization method out of Group AbsMax, AWQ, OmniQuant, and AffineQuant. For completeness, we have provided the accuracy achieved by each of these quantization methods separately in Table 15.

Methods like OmniQuant and AffineQuant encounter difficulties in quantizing OPT-350M, resulting in NaN values. Additionally, approaches such as AWQ, OmniQuant, and AffineQuant cause memory issues (OOM) when attempting to compress the models on a single A100-40GB GPU.

*Table 15.* Average zero-shot accuracy of LLaMA-2 and OPT models with **2:4 sparsity and 4-bit weight quantization**. ↑ indicates better performance.

| Pruning/LoRA Method | Weight Quantization | OPT | | | | | | LLaMA-2 | |
|---|---|---|---|---|---|---|---|---|---|
| | | 125M | 350M | 1.3B | 2.7B | 6.7B | 13B | 7B | 13B |
| Dense | - | 35.9 | 37.1 | 43.4 | 45.5 | 48.3 | 48.7 | 56.6 | 60.8 |
| **2:4 Sparsity** | | | | | | | | | |
| Wanda | Group AbsMax | 33.27 | 32.79 | 37.47 | 39.45 | 42.95 | 43.64 | 43.89 | 48.94 |
| Wanda | AWQ | 33.33 | 31.50 | 38.43 | 40.00 | 43.41 | 44.07 | 44.86 | OOM |
| Wanda | OmniQuant | 33.37 | NaN | 37.35 | 39.39 | 41.50 | OOM | 43.95 | OOM |
| Wanda | AffineQuant | 33.39 | NaN | 37.48 | 33.51 | 42.88 | OOM | 44.62 | OOM |
| **50% Unstructured** | | | | | | | | | |
| Wanda | Group AbsMax | 34.67 | 33.89 | 40.38 | 42.77 | 45.88 | 46.60 | 51.76 | 56.76 |
| Wanda | AWQ | 35.11 | 31.57 | 41.02 | 42.89 | 46.52 | 46.84 | 50.68 | OOM |
| Wanda | OmniQuant | 34.85 | NaN | 39.84 | 42.16 | 44.67 | OOM | 50.51 | OOM |
| Wanda | AffineQuant | 34.64 | NaN | 41.23 | 42.68 | 46.05 | OOM | 53.62 | OOM |

## I. Sparsity vs. Quantization

A natural question that arises compressing models is whether it is more efficient to reduce the model size through pruning or quantization. To answer this question, we conduct a set of experiments, which evaluate the perplexity of different models under three different conditions, all with around $8\times$ model size reduction factor: (1) 2-bit weight quantization with no sparsity, (2) 4-bit weight quantization with 50% unstructured sparsity, and (3) 4-bit weight quantization with 50% 2:4 sparsity. We have used SLIM-LoRA with SLIM-Quant in all the experiments. The accuracy and perplexity results of these experiments are summarized in Tables 16 and 17, showing that combining sparsity and quantization yields better results in comparison to quantization-only settings with lower bitwidth.

*Table 16.* Average accuracy of different models on WikiText-2 dataset using different pruning and quantization schemes. ↑ indicates better performance. Combining sparsity and quantization provides better accuracy results in comparison to solely using quantization.

| Quantization | Sparsity | OPT | | | | | | LLaMA-2 | |
|---|---|---|---|---|---|---|---|---|---|
| | | 125M | 350M | 1.3B | 2.7B | 6.7B | 13B | 7B | 13B |
| 2-bit | - | 33.5 | 32.5 | 38.5 | 39.2 | 43.8 | 44.4 | 42.4 | 44.9 |
| 4-bit | 2:4 | 34.6 | 34.4 | 40.6 | 42.7 | 46.0 | 46.1 | 51.2 | 54.9 |
| 4-bit | 50% Unstructured | 35.2 | 35.3 | 41.9 | 43.5 | 47.1 | 48.0 | 54.3 | 57.9 |

*Table 17.* Perplexity of different models on WikiText-2 dataset using different pruning and quantization schemes. ↓ indicates better performance. Combining sparsity and quantization provides better accuracy results in comparison to solely using quantization.

| Quantization | Sparsity | OPT | | | | | | LLaMA-2 | |
|---|---|---|---|---|---|---|---|---|---|
| | | 125M | 350M | 1.3B | 2.7B | 6.7B | 13B | 7B | 13B |
| 2-bit | - | 116.2 | 169.7 | 35.1 | 27.1 | 16.2 | 15.0 | 12.5 | 11.7 |
| 4-bit | 2:4 | 47.5 | 45.6 | 18.8 | 15.7 | 12.4 | 12.1 | 7.2 | 6.5 |
| 4-bit | 50% Unstructured | 36.3 | 29.9 | 16.3 | 13.7 | 11.4 | 10.8 | 6.0 | 5.4 |

## J. Additional Speedup Results

Section 4 presents the speedup of SLɪM on consumer-grade GPUs, while this section provides results on NVIDIA A100-40GB GPUs. Figure 4 summarizes the speedup for the LLaMA-2 and LLaMA-3.1 model families, including LLaMA-3.1-405B, highlighting SLɪM ' scalability to large models. As with consumer-grade devices, larger models achieve higher speedups.

Additionally, the breakdown of the speedup, showing the contribution of quantization and sparsity, is demonstrated using brighter and darker colors, respectively.

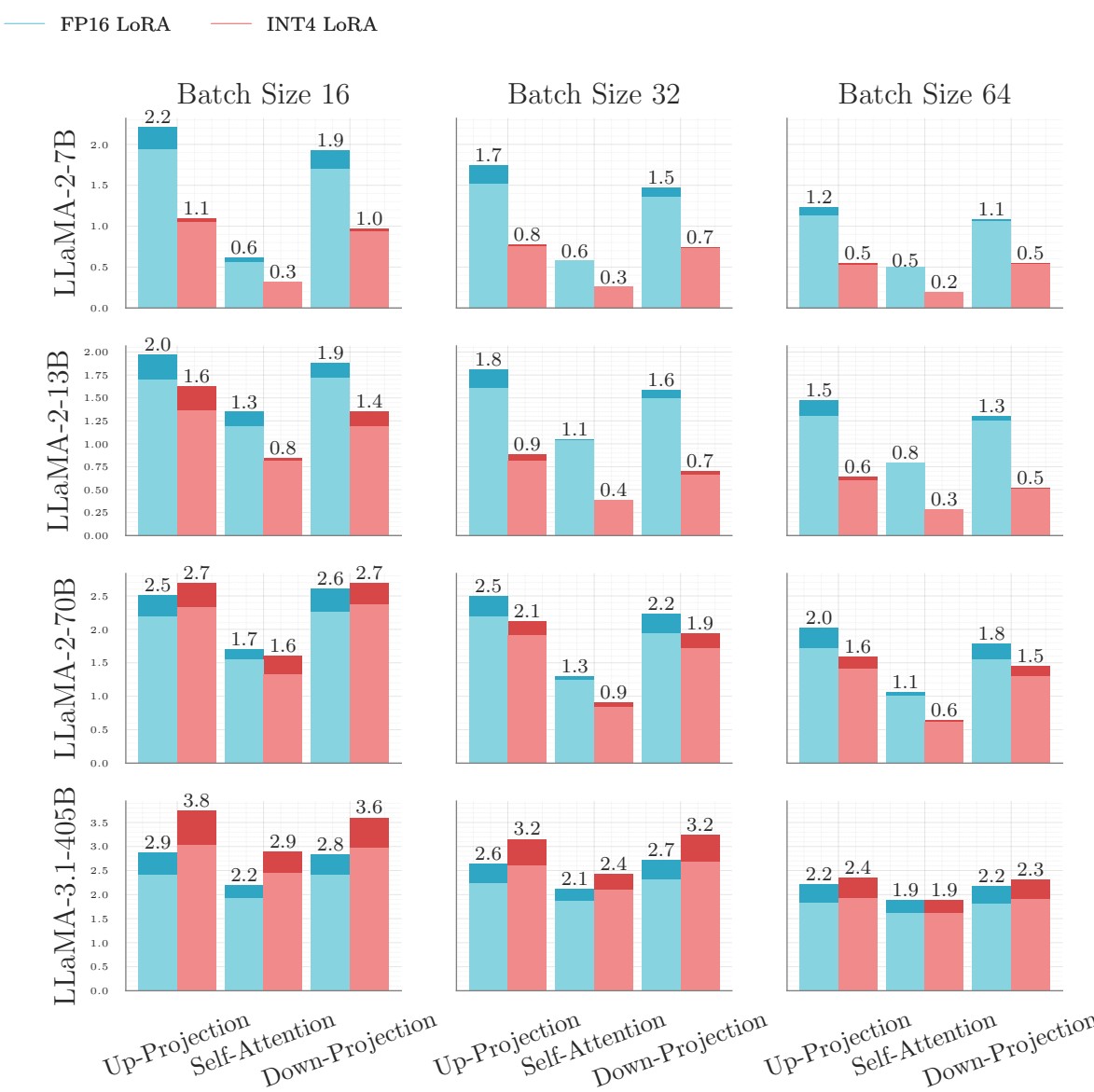

Figure 4. SLɪM speedup for LLaMA-2 family of models on NVIDIA A100-40GB GPUs. The brighter color shows the contribution of quantization to the total speedup.

## K. Fine-tuning Costs

Fine-tuning compressed models can recover lost accuracy, but the high parameter count leads to substantial time and memory costs. In our experiments, we fine-tuned models with low-rank adapters, where the quantized weights are frozen and only the adapters are fine-tuned. This results in a more parameter-efficient approach, reducing both memory and computational costs. When no low-rank adapter is used, the straight-through estimator (STE) fine-tunes the quantized weights.

Table 18 presents the fine-tuning results for 300,000 tokens from the C4 dataset, using a batch size of 64 and sequence length of 1024 on a single H100 GPU. Fine-tuning models without low-rank adapters took 12 hours for 125M parameter models and over 36 days for 13B parameter models. Given these high costs, completing fine-tuning was challenging with our limited resources. In contrast, using low-rank adapters and freezing the sparse quantized weights made fine-tuning more efficient, enabling us to report accuracy results in Table 1.

*Table 18.* The required time for fine-tuning the models with a single H100 GPU on 300,000 tokens from the C4 dataset with a batch size of 64 and a sequence length of 1024.

| Pruning Method | Weight Quantization | OPT | | | | | | LLaMA-2 | |
|---|---|---|---|---|---|---|---|---|---|
| | | 125M | 350M | 1.3B | 2.7B | 6.7B | 13B | 7B | 13B |
| Magnitude | Group AbsMax | | | | | | | | |
| SparseGPT | OPTQ | 12h | 43h | 164h | 361h | 866h | 867h | 842h | 844h |
| Wanda | Group AbsMax | | | | | | | | |
| SLɪM-Naive | SLɪM-Quant$^W$ | 1.5h | 3h | 6h | 8h | 16h | 18h | 14h | 14h |
| SLɪM-LoRA | SLɪM-Quant$^W$ | | | | | | | | |

## L. Memory Reduction Analysis

SLɪM prunes and quantizes the models and adds additional low-rank adapters to them. Additionally, it supports quantization methods for the low-rank adapters to reduce their overhead. In the following, we propose an analysis of the reduced memory when using SLɪM and other pruning and quantization methods.

Assuming the hidden dimension of a model is $d$ and the low-rank adapter ratio used in the model is of rank $r < 1$. Furthermore, by denoting the number of transformer blocks with $n$ and the vocabulary size of the model by $V$ and by denoting the ratio of the up-projection and down-projection layers in the model by $a$, we can get the memory reduction as the ratio of $\frac{\text{Compressed Model Size}}{\text{Dense Model Size}}$ from equation 12.

$$\text{Memory Reduction} = \frac{n(4d^2 + 2d^2a) + dV}{n(4d^2/2 + 4 \times 2d^2r + 2d^2a/2 + 2d(dr + dra)) + dV} \tag{12}$$

Table 19 summarizes the memory reduction of different pruning and quantization methods. Please note that when using low-rank adapters (in Naive-LoRA and SLɪM-LoRA), we assume a rank of $r = 0.1$.

*Table 19.* Theoretical memory reduction ($\times$) of different compression methods across various OPT and LLaMA models. In Quantized SLɪM , the low-rank adapters are also quantized.($\downarrow$ indicates better performance.)

| Compression Method | OPT | | | | | | LLaMA-2 | |
|---|---|---|---|---|---|---|---|---|
| | 125M | 350M | 1.3B | 2.7B | 6.7B | 13B | 7B | 13B |
| SparseGPT + OPTQ | 0.40 | 0.30 | 0.25 | 0.17 | 0.15 | 0.14 | 0.15 | 0.14 |
| Wanda + AbsMax | 0.40 | 0.30 | 0.25 | 0.17 | 0.15 | 0.14 | 0.15 | 0.14 |
| Naive-LoRA + AbsMax | 0.50 | 0.42 | 0.38 | 0.31 | 0.30 | 0.29 | 0.31 | 0.30 |
| SLɪM-LoRA + SLɪM-Quant | 0.50 | 0.42 | 0.38 | 0.31 | 0.30 | 0.29 | 0.31 | 0.30 |
| SLɪM-LoRA$^Q$ + SLɪM-Quant | 0.42 | 0.33 | 0.28 | 0.20 | 0.19 | 0.18 | 0.19 | 0.18 |

## M. Computation Reduction Analysis

SLIM and other compression methods reduce the number of floating point operations (FLOPs) at the inference of models. Additionally, the low-rank adapters used in SLIM and Wanda SVD can add additional computational overheads to the inference of the models. Following JSQ (Guo et al., 2024), in this section, we provide an analysis on the FLOP reduction in the inference of different methods. It is noteworthy that even though quantization can reduce the memory overhead of models, since all the computations are done in floating point format, it does not lead to a reduction in the computation of the inference.

Assuming the hidden dimension of a model is $d$ and the low-rank adapter ratio used in the model is of rank $r < 1$. Furthermore, by denoting the number of transformer blocks with $n$ and the vocabulary size of the model by $V$ and by denoting the ratio of the up-projection and down-projection layers in the model by $a$, we can get the memory reduction as the ratio of $\frac{\text{Dense Inference FLOP Count}}{\text{Compressed Inference FLOP Count}}$ from equation 13, where $b$ is the batch size, and is canceled in the numerator and the denominator of the equation.

$$\text{FLOP Reduction} = \frac{n(4bd^2 + 2bd^2a) + bdV}{n(4bd^2/2 + 4 \times 2bd^2r + 2bd^2a/2 + 2b(d^2r + d^2ra)) + bdV} \tag{13}$$

Table 20 summarizes the FLOP reduction of different compression methods. As it can be seen, the overhead of adding the low-rank adapters ($r = 0.1$) in SLIM-LoRA and Naive-LoRA is not significant.

*Table 20.* Compute (FLOP) reduction ratios ($\times$) of different compression methods across various OPT and LLaMA models. In Quantized SLIM , the low-rank adapters are also quantized. ($\uparrow$ indicates better performance.)

| Compression | OPT | | | | | | LLaMA-2 | |
|---|---|---|---|---|---|---|---|---|
| Method | 125M | 350M | 1.3B | 2.7B | 6.7B | 13B | 7B | 13B |
| SparseGPT + OPTQ | 1.52 | 1.66 | 1.75 | 1.91 | 1.94 | 1.96 | 1.95 | 1.97 |
| Wanda + AbsMax | 1.52 | 1.66 | 1.75 | 1.91 | 1.94 | 1.96 | 1.95 | 1.97 |
| Naive-LoRA + AbsMax | 1.32 | 1.39 | 1.43 | 1.50 | 1.51 | 1.52 | 1.49 | 1.49 |
| SLIM-LoRA + SLIM-Quant | 1.32 | 1.39 | 1.43 | 1.50 | 1.51 | 1.52 | 1.49 | 1.49 |
| SLIM-LoRA$^Q$ + SLIM-Quant | 1.32 | 1.39 | 1.43 | 1.50 | 1.51 | 1.52 | 1.49 | 1.49 |

## N. Compression Costs

The computational cost of compression methods varies depending on their complexity. While all approaches can compress a single layer at a time, the memory usage is similar across methods, as each stores only one layer in the GPU's global memory. Techniques like Wanda, which rely on matrix multiplication, are faster than more complex methods like SparseGPT, which computes the inverse Hessian matrix for each layer. Adding low-rank adapters to Wanda-SVD and SLIM increases computational complexity due to the need for singular value decomposition (SVD), making them comparable to SparseGPT in terms of computation.

Table 21 summarizes the time required to compress various models using the discussed methods. Methods incorporating low-rank adapters (SLIM and Wanda-SVD) generally take longer to compress due to their higher complexity. Interestingly, SparseGPT's compression time is comparable to methods with low-rank adapters, despite only performing pruning and quantization. The saliency-based approach in SLIM does not add significant overhead compared to Wanda-SVD, maintaining efficiency despite its added complexity.

*Table 21.* The required compresion time for different models and compression methods using a single H100 GPU.

| Pruning Method | Weight Quantization | OPT | | | | | | LLaMA-2 | |
|---|---|---|---|---|---|---|---|---|---|
| | | 125M | 350M | 1.3B | 2.7B | 6.7B | 13B | 7B | 13B |
| Magnitude | AbsMax | 1s | 1s | 1s | 1s | 2s | 4s | 2s | 4s |
| SparseGPT | OPTQ | 1m | 2m | 5m | 11m | 22m | 41m | 25m | 46m |
| Wanda | SLIM-Quant | 0.5m | 1m | 3m | 5m | 8m | 13m | 8m | 14m |
| Wanda-SVD | SLIM-Quant | 1m | 2m | 7m | 13m | 33m | 60m | 38m | 67m |
| SLIM | SLIM-Quant | 1m | 2m | 7m | 13m | 34m | 63m | 39m | 68m |

## O. Rank Analysis

The key hyperparameter in low-rank approximation is the rank of the adapters. While increasing the rank reduces approximation error, it also leads to higher computational and memory overhead. Therefore, it is crucial to analyze the trade-off between the accuracy improvements and the overhead introduced by the chosen approximation rank.

Assuming the rank of the low-rank adapter is $rd$, where $r < 1$ is a fixed factor and $d$ is the dimension of the weights in a square feed-forward layer, the low-rank adapters are represented as $\mathcal{L}, \mathcal{R}^T \in \mathbb{R}^{d \times rd}$, resulting in a memory overhead of $\mathcal{O}(2rd^2)$ for storing them. To compute $\mathcal{X}\mathcal{L}\mathcal{R}$, where $\mathcal{X} \in \mathbb{R}^{b \times d}$ is the input with a batch size of $b$, the computational complexity is $\mathcal{O}(2brd^2)$. Given that the original memory and computational complexity of the layer are $\mathcal{O}(d^2)$ and $\mathcal{O}(bd^2)$, respectively, the overhead introduced by the low-rank adapters becomes negligible when $r \ll 1$.

Figure O-a shows the average zero-shot accuracy of the OPT-6.7B and LLaMA-2-7B models for various ranks. As expected, increasing the rank leads to improved model accuracy. Based on these results, a rank of $r = 0.1$ provides a substantial boost in accuracy without introducing significant overhead to inference.

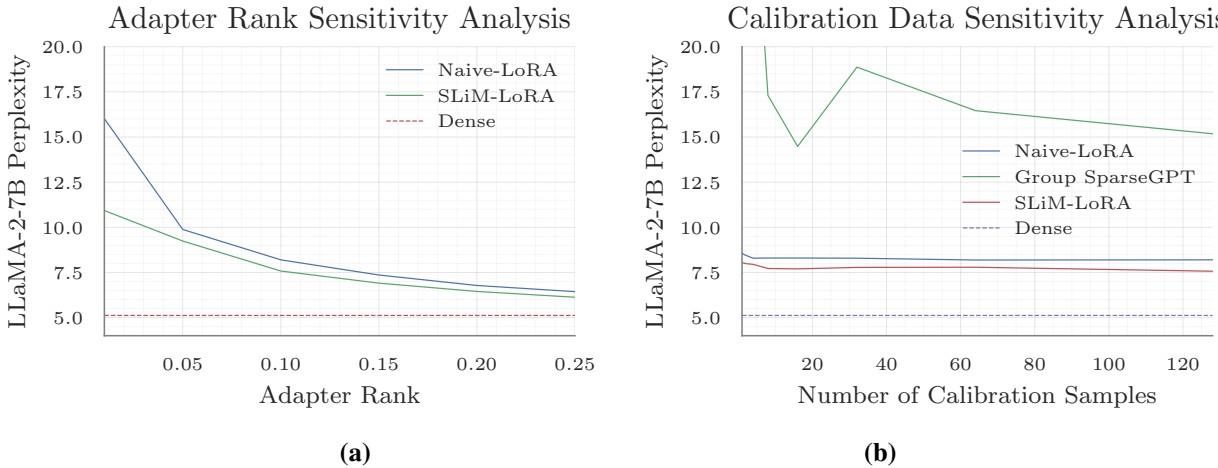

**(a)** **(b)**

*Figure 5.* Sensitivity analysis for the rank of the adapter (**a**) and the number of calibration samples (**b**) for different one-shot compression methods. For Naive-LoRA and SLIM-LoRA, we have used the SLIM-Quant quantization method, and for the SparseGPT, we have used the Group quantization version of OPTQ.

## P. Effects of Calibration Sample Count

Similar to previous work (SparseGPT, Wanda, AWQ, OmniQuant, and AffineQuant), SLIM leverages a set of calibration data from the C4 dataset to assess weight saliency for pruning and low-rank approximations. Figure O-b illustrates the perplexity of LLaMA-2-7B using varying numbers of calibration samples. As shown, SLIM demonstrates low sensitivity to the number of calibration samples, making it effective even in scenarios with limited data.

## Q. Sensitivity to Calibration Dataset

Similar to other pruning and quantization methods such as Wanda, SparseGPT, OPTQ, and AWQ, SLɪM relies on a calibration dataset to evaluate weight saliency. The C4 (Raffel et al., 2019) and SlimPajama (Soboleva et al., 2023) datasets are among the most commonly used calibration sets for LLM compression. Table 22 presents the perplexity results for SLɪM-LoRA and SLɪM-Quant across different calibration datasets. The results indicate that SLɪM is largely insensitive to the choice of dataset, achieving comparable accuracy regardless of the calibration dataset used.

*Table 22.* Perplexity of different models on WikiText-2 dataset using SLɪM-LoRA with 4-bit quantization using SLɪM-Quant with different calibration datasets. ↓ indicates better performance.

| Calibration | OPT | | | | | | LLaMA-2 | |
| Dataset | 125M | 350M | 1.3B | 2.7B | 6.7B | 13B | 7B | 13B |
|---|---|---|---|---|---|---|---|---|
| **50% 2:4** | | | | | | | | |
| C4 | 57.91 | 50.09 | 19.64 | 15.65 | 12.71 | 12.13 | 7.56 | 6.50 |
| SlimPajama | 46.27 | 44.77 | 19.35 | 16.04 | 12.56 | 12.32 | 7.15 | 6.49 |
| **50% Unstructured** | | | | | | | | |
| C4 | 39.62 | 31.51 | 16.52 | 13.65 | 11.42 | 10.82 | 6.16 | 5.36 |
| SlimPajama | 36.49 | 29.94 | 16.64 | 14.08 | 11.61 | 11.02 | 5.99 | 5.34 |

## R. Sparsity Analysis

To analyze the impact of sparsity on model accuracy, we conduct experiments on LLaMA-2-13B with 4-bit quantization, pruning it to varying sparsity ratios. Figure 6 presents the perplexity results for SLɪM-LoRA with SLɪM-Quant , SparseGPT with OPTQ, and Wanda with Group AbsMax. As expected, increasing the sparsity ratio leads to higher perplexity, indicating a trade-off between compression and accuracy. Notably, SLɪM-LoRA combined with SLɪM-Quant maintains competitive accuracy up to 60% sparsity, whereas other methods experience noticeable degradation at lower sparsity levels.

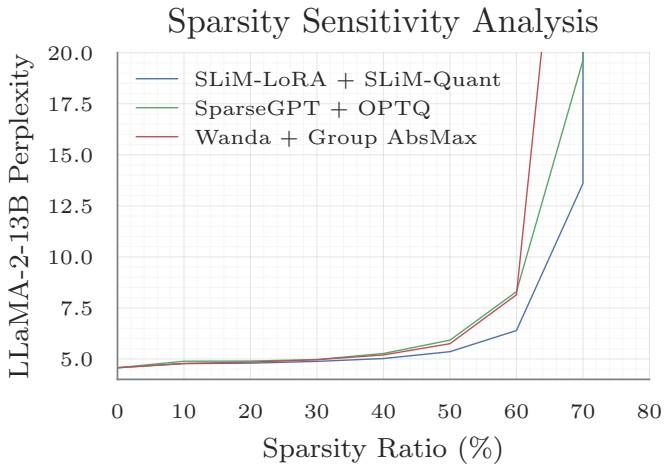

*Figure 6.* Sparsity analysis on LLaMA-2-13B model using perplexity on WikiText-2 dataset. ↓ indicates better performance.

## S. Related Work

SLɪM combines model pruning and quantization for compression, complemented by zero-shot low-rank adapters to recover lost accuracy. This section reviews related work on these topics.

## S.1. Pruning

Eliminating redundant weights reduces computation and memory costs during inference. Optimal Brain Damage (OBD) (LeCun et al., 1989) leverages second-order information of the loss function to identify the least important weights but is computationally prohibitive for large language models (LLMs) (Mozaffari et al., 2023). WoodFisher (Singh & Alistarh, 2020) approximates the Hessian matrix using Kronecker Factorization to mitigate this overhead but struggles to scale to LLMs.

Optimal Brain Surgeon (OBS) (Hassibi et al., 1993) evaluates weight matrices layer-wise using the layer-wise Hessian matrix to preserve layer outputs. However, the cubic growth in the cost of inverting the layer-wise Hessian with model size renders this approach impractical for LLMs. Optimal Brain Compression (OBC) (Frantar & Alistarh, 2022) addresses the OBS-defined compression problem using a greedy algorithm, while SparseGPT reformulates it as a sparse regression problem. Wanda introduces a lightweight method based on weight and activation magnitudes to identify unimportant weights without updating their values.

In addition to post-training sparsity, a recent line of work targets sparsity during training (Lu et al., 2023; Mozaffari et al., 2024; Bambhaniya et al., 2024); however, their applicability is limited because of the expensive costs of training.

## S.2. Quantization

Quantizing all elements in a matrix is challenging due to the significant impact of outliers on the model (Dettmers et al., 2022). Group quantization (Alistarh et al., 2017; Gunho et al., 2022) addresses this by quantizing small groups of a weight matrix with a shared quantization parameter, but it introduces challenges discussed in Appendix U.

AbsMax (Jacob et al., 2018) with round-to-nearest (RTN) is the simplest quantization scheme for matrix elements. OPTQ (Frantar et al., 2022) minimizes layer-wise error using an approach akin to OBS. AWQ (Lin et al., 2024) shifts the challenge of quantizing salient weights to activations, while SmoothQuant (Xiao et al., 2023) balances quantization error between weights and activations, enabling input quantization. OmniQuant (Shao et al., 2023) improves accuracy with learnable clipping and channel scaling. AffineQuant leverages equivalent affine transformations to reduce quantization error, and QuaRot (Ashkboos et al., 2024) uses rotations to eliminate outliers during quantization.

Advanced methods like JSQ (Guo et al., 2024) jointly prune and quantize weights to 8 bits but struggle to recover accuracy in low bit-width quantization, limiting their utility. An analysis of the interplay between sparsity and quantization can be found in (Harma et al., 2024).

## S.3. Low-rank Adapters

Low-rank adapters were first introduced to LLMs to reduce the overhead of fine-tuning (Hu et al., 2021; Mozaffari et al., 2024). Q-LoRA (Dettmers et al., 2023) extended this approach by quantizing weights before fine-tuning, allowing the process to recover accuracy lost during quantization. LQ-LoRA (Guo et al., 2023) further improved Q-LoRA by initializing the adapters using the SVD of the quantization error. LoSparse (Li et al., 2023) has a similar approach as LQ-LoRA, but for sparsity, initializing the low-rank adapters to the norm of the pruning error. RoSA (Nikdan et al., 2024) expands the learning capability of the model by adding both low-rank and sparse adapters to the model. This approach adds an extra sparse matrix multiplication to the inference, increasing the adapter overhead even further. However, all these methods require hundreds of millions of tokens for fine-tuning, making them costly and not comparable to one-shot pruning and quantization methods, or methods that use much shorter fine-tuning phases.

$L^2$QER (Zhang et al., 2024a) avoids fine-tuning by using one-shot low-rank adapters to mitigate quantization error. However, it performs poorly when combined with sparsity, resulting in a significant accuracy gap between the compressed and dense models. More recent methods, QERA (Zhang et al., 2024b) and CALDERA (Saha et al., 2024) find closed-form solutions to the problem discussed in $L^2$QER, but they still do not support sparsity.

# T. Settings and Hyperparameters

To ensure a fair comparison and robust performance, SLIM utilizes calibration data and fine-tuning datasets under the same conditions as leading one-shot pruning and quantization methods. Similar to Wanda, SparseGPT, and OPTQ, SLIM leverages calibration data to extract statistics and assess weight saliency. Specifically, we use 128 sequences sampled from the widely-

used C4 (Raffel et al., 2019) dataset for calibration. Additionally, for all fine-tuning experiments, we employ 300,000 tokens from the C4 dataset to improve model accuracy post-compression. This standardized approach to data usage ensures that SLıM operates under the same conditions as its peers, enabling a fair evaluation of its compression and fine-tuning performance.

SLıM-Quant uses the histogram of the weight elements to find the optimal scaling factor. The use of the histogram reduces the overhead of finding the optimal parameter by sharing the error computations between the elements that fall into the same histogram bin. The number of histogram bins provides a trade-off between the computational overhead and the accuracy of the integration. We set the number of bins in the histogram to $\max(512, \min(\frac{d_{in} \times d_{out}}{1000}, 20,000))$ to achieve an accurate approximation of the PDF of the data.

We standardize our experimental setup by detailing the quantization scheme, group quantization parameters, and low-rank adapter configurations to ensure reproducibility and comparability across methods. All quantization methods in the experiments follow a 4-bit weight-only quantization scheme. Consistent with prior work (OPTQ, OmniQuant, AffineQuant, etc.), group quantization uses a group size of 128. For experiments involving Naive-LoRA and SLıM-LoRA, we set the adapter rank to 10% of the model's hidden dimension unless stated otherwise. These standardized configurations ensure consistency with prior work and enable a fair comparison of SLıM against baseline methods.

For fine-tuning the models, we utilized the Hugging Face Trainer (Wolf, 2019). The AdaFactor (Shazeer & Stern, 2018) optimizer was employed during the fine-tuning process, accompanied by linear learning rate scheduling. The optimization and learning rate scheduling parameters were set to their default values in the Hugging Face Trainer. To prevent numerical overflow and divergence, we used BFloat-16 data types (Wang & Kanwar, 2019) available on NVIDIA A100 GPUs during fine-tuning. The training was conducted with a local batch size of 1 and a gradient accumulation factor of 64 to reduce memory overhead. Weight updates for the sparse and/or quantized weights, as well as the corresponding biases, were disabled. Due to our limited resources, we did not tune any of the hyperparameters aimed at improving fine-tuning speed or accuracy; tuning these parameters is planned for future work.

## U. Group Quantization Challenges

Group quantization allows sharing the same quantization parameters for a small group of the elements in the quantized matrix, leading to smaller errors. But, using group quantization adds additional challenges to the training and inference of the model, e.g. **more complicated implementation** and **additional memory and compute overheads**.

The state-of-the-art group quantization GPU kernel, dense and sparse Marlin (Frantar et al., 2024), consists of thousands of lines of CUDA code optimized for only a limited number of GPU architectures, showcasing the amount of effort needed to implement a version of group quantization. Furthermore, other libraries and frameworks, such as Triton (Tillet et al., 2019) and CUTLASS (NVIDIA Corporation, 2025) do not provide support for 4-bit group quantization, limiting its flexibility and possibility of modification.

Furthermore, using group quantization can lead to an additional overhead during matrix multiplication, since more parameters need to be loaded for dequantizing each group. As an example, Table 23 shows the slow-down of using group quantization on the down-projection matrices in different LLaMA-2 and LLaMA-3.1 models on a NVIDIA A100-40GB GPU, with a batch size of 16.

Table 23. Group quantization slow-down ($\times$) on different LLaMA-2 and LLaMA 3.1 models. $\downarrow$ indicates worse.

| Model | LLaMA-2-7B | LLaMA-2-13B | LLaMA-2-70B | LLaMA-3.1-405B |
|---|---|---|---|---|
| Slow-Down ($\times$) | 0.94 | 0.95 | 0.95 | 0.94 |

