# OpenReview forum: "SLiM: One-shot Quantization and Sparsity with Low-rank Approximation for LLM Weight Compression"
_ICML.cc/2025/Conference — ICML 2025 poster_

### Official Review · Reviewer_UKqS · 2025-03-11

**Overall Recommendation:** 3

**Summary:**

Large language models contain extensive parameter counts leading to significant memory overhead and high inference costs. Pruning and quantization methods solve this, but typically both need retraining on large-scale datasets. One-shot methods can reduce the cost, but jointly pruning and quantizing weights under low-bit scenarios is a challenge. This study proposes SLIM, which combines one-shot quantization, low-rank adapters and sparsity to compress language models for efficient inference. SLIM is validated on OPT and LLaMA-2 families, achieving significant improvements in model efficiency and accuracy across various benchmarks.

This study highlights the following:

- **SLIM-Quant**: SLIM adopts symmetric weight quantization, and clips the weights before rounding to the nearest integer. The clipping threshold is a hyperparameter, and SLIM-Quant searches this parameter by transferring the problem into a probabilistic formulation of the quantization process. The authors finally use grid search to tackle the optimal $\alpha$.
- **SLIM-LoRA**: Naive-LoRA is a straightforward approach that minimizes the total error norm between the original weight matrix and the compressed weight matrix, but it overlooks the importance of each weight value. SLIM-LoRA takes $\operatorname{diag}(\mathbf{x})$ as a saliency function F, and minimizes $F(E_Q)$. The solution to this is to compute the SVD of $F (−(E_Q + E_S))$.
- **Low-rank adapter quantization**: The authors further quantize the low-rank adapters to save memory and computation. They use AbsMax group quantization scheme for the adapters and group size is 128.
- **Post-compression fine-tuning**: The authors only fine-tune the adapters for efficiency. Straight-through estimator (STE) method is used in back propagation.

SLIM is evaluated on the OPT and LLaMA-2 model families. Downstream tasks include MMLU, Piqa, Arc-Easy, Arc-Challenge, WinoGrande and OpenBookQA. The authors also report PPL on WikiText2. Baseline methods are one-shot pruning methods like Wanda and SparseGPT, L$^2$QER, JSQ. Compared with those methods, SLIM show advantage both on structured and unstructured sparsity. The authors also report acceleration results, on RTX 3060 and A100 GPUs. Acceleration ratio is about 2~3x compared with dense full precision models.

**Claims And Evidence:**

Most of the claims are clear and convincing.

**Essential References Not Discussed:**

No missing related works.

**Experimental Designs Or Analyses:**

Most of the experimental designs are valid.

**Methods And Evaluation Criteria:**

Most of the methods and evaluation criteria make sense.

**Other Comments Or Suggestions:**

Typos:

1. Extra space on line 218.

**Other Strengths And Weaknesses:**

Strengths:

1. The paper is well organized and well written. The equations are very clear and easy to understand. The technical content is explained in sufficient detail. Additionally, the use of figures, tables, and examples enhances the clarity of the presentation, ensuring that the key contributions and findings are easy to follow and understand.
2. Combining sparsity, quantization and low-rank adapters is challenging. However, the authors do this successfully.

Weaknesses:

1. The improvement does not seem significant. I wonder in practical instruction-following cases, such as MATH or GSM8K, would the advantage still exist.

**Questions For Authors:**

Questions:

1. Regarding the acceleration in Table 3, is the acceleration mainly from quantization, or is it from sparsity? It would be better to provide an ablation study.
2. Since 'one-shot' is mentioned in the title, what are the results if the post-compression fine-tuning stage is omitted? An ablation study on this would be beneficial.

Overall, the paper demonstrates strong clarity and structure. I appreciate the authors' efforts in translating these techniques into practical applications. I recommend a weak accept and suggest that the authors address W1, Q1 and Q2 to provide a more comprehensive perspective. The scores may be revised following the author-reviewer discussion.

**Relation To Broader Scientific Literature:**

This study can be seen as an extension or improvement for the N:M post-training pruning. This study is closely related to previous N:M pruning works, such as SparseGPT and Wanda.

**Theoretical Claims:**

As I check, this article is based on experiments and does not require strict theoretical proof.

---

> ### Author Rebuttal · Authors · 2025-04-01
>
> We thank the reviewer for their helpful comments. We have provided a detailed reply to address all of your points.
>
> # Significance of Accuracy Improvements and MATH Benchmark
> SLiM achieves up to 5.66% higher average accuracy than leading compression methods across six zero-shot tasks.
> Per your request, we evaluated LLaMA-2-7B/13B on the MATH benchmark using default LM-Evaluation Harness settings.
> Note that our vanilla models do not employ CoT[1], which can boost math problem performance. Evaluations on GSM8K are ongoing but slow (over 20 hours for a 13B model). We are also testing quantization on sparse checkpoints and will share results promptly, despite rebuttal constraints.
>
> ## LLaMA-2 7B
>
> \\begin{array}{|c|c|c|}
> \\hline
> \\rowcolor[gray]{0.9} Method & Sparsity Pattern & MATH \\\\
> \\hline
> \\rowcolor[gray]{0.95} Dense & N/A & 0.26 \\\\
> \\hline
> \\rowcolor[gray]{1.0} SparseGPT + OPTQ & 2:4 & 0.04 \\\\
> \\rowcolor[gray]{0.95} Wanda + Group AbsMax & 2:4 & 0.00 \\\\
> \\rowcolor[gray]{1.0} SLiM-LoRA + SLiM-Quant & 2:4 & 0.64 \\\\
> \\hline
> \\rowcolor[gray]{0.95} SparseGPT + OPTQ & Unstructured & 0.18 \\\\
> \\rowcolor[gray]{1.0} Wanda + Group AbsMax & Unstructured & 0.04 \\\\
> \\rowcolor[gray]{0.95} SLiM-LoRA + SLiM-Quant & Unstructured & 0.34 \\\\
> \\hline
> \\end{array}
>
> ## LLaMA-2 13B
>
> \\begin{array}{|c|c|c|c|}
> \\hline
> \\rowcolor[gray]{0.9} Method & Sparsity Pattern & MATH \\\\
> \\hline
> \\rowcolor[gray]{0.95} Dense & N/A & 0.60 \\\\
> \\hline
> \\rowcolor[gray]{1.0} SparseGPT + OPTQ & 2:4 & 0.34 \\\\
> \\rowcolor[gray]{0.95} Wanda + Group AbsMax & 2:4 & 0.12 \\\\
> \\rowcolor[gray]{1.0} SLiM-LoRA + SLiM-Quant & 2:4 & 0.62 \\\\
> \\hline
> \\rowcolor[gray]{0.95} SparseGPT + OPTQ & Unstructured & 0.46 \\\\
> \\rowcolor[gray]{1.0} Wanda + Group AbsMax & Unstructured & 0.36 \\\\
> \\rowcolor[gray]{0.95} SLiM-LoRA + SLiM-Quant & Unstructured & 0.69 \\\\
> \\hline
> \\end{array}
>
> # Speedup Breakdown
>
> We’ve included [this graph](bit.ly/3QXcit2), showing the contributions of quantization and sparsity to SLiM’s speedup, evaluated in Quantized-only and Sparse+Quantized settings using Sparse Marling kernels in vLLM. The results indicate that quantization drives most of the speedup, with sparsity contributing less.
>
> # Optionality of Post-Compression Fine-Tuning
>
> SLiM is a one-shot compression method that improves model accuracy without requiring fine-tuning. The results in `Table-1-Page-7` demonstrate that SLiM surpasses state-of-the-art compression methods without any fine-tuning.
>
> While an **optional** PEFT step can further enhance accuracy, it is not integral to SLiM’s core approach. Additional improvements from PEFT are detailed in the ablation studies in `Table-2-Page-7` and `Table-7-Page-14`.
>
> [1] Wei, et al. “Chain-of-Thought Prompting Elicits Reasoning in Large Language Models”,  NeurIPS 2022

---

> > ### Comment · Reviewer_UKqS · 2025-04-05
> >
> > Thank you for answering my questions. Most of my concerns are addressed, except for the speedup breakdown graph which does not show on OpenReview. The experiments are comprehensive and ideas are easy but effective. I will recommend a weak accept for your paper.

---

> > > ### Author Response · Authors · 2025-04-07
> > >
> > > Dear Reviewer UKqS,
> > >
> > > We sincerely thank you for your positive feedback on our submission. We apologize for the inconvenience caused by the broken link to the speedup breakdown graph in our previous rebuttal. This issue has been resolved, and the graph is now accessible [here](https://github.com/anonymous-m13/slim-icml2025/blob/main/rtx3060_speedup.pdf). We are more than happy to address any additional questions or concerns you may have.
> > >
> > > Best Regards,
> > > Authors

---

### Official Review · Reviewer_q6Mp · 2025-03-13

**Overall Recommendation:** 3

**Summary:**

This paper proposes to use low-rank approximation to reduce the compression error for quantization and pruning on LLM. SLIM-Quant minimizes the quantization error by selecting the optimal scaling parameter. Low-rank adapters are applied to compensate the quantization and pruning error. Quantization and fine-tuning on the adapter further improve the SLIM. The results show the model performance improvement for 4 bits+50% sparsity compressed model.

**Claims And Evidence:**

Yes. It claims improvement on model accuracy and the speedup on RTX3060 and A100 GPUs. The results show the improvement on different models and tasks, which verify the claim.

**Essential References Not Discussed:**

No.

**Experimental Designs Or Analyses:**

I keep up with the literature in this area.

**Methods And Evaluation Criteria:**

Yes. The proposed methods can effectively improve the quantized and pruned model performance by using low-rank adapters. But the evaluation for speedup performance should be compared with other quantization and pruning tasks besides FP16 model.

**Other Comments Or Suggestions:**

N/A

**Other Strengths And Weaknesses:**

Strength:
- The proposed methods have good performance on pruned and quantized model.
- The Additional Experiments are comprehensive.
Weakness:
- The speedup for LLM mainly benefit from the quantization, pruning architecture and the hardware. So, the speedup performance comparison is not fair for the proposed methods. It should be compared with structure pruned model and quantized model. The SLIM-LoRA may reduce the inference performance because of additional Low-Rank operation.
- Although the performance is good than other quantized and pruned method, it still has much loss compared with dense model. For Llama2-13B, the SLIM-LoRA compressed model cause 5.86% accuracy loss for zero-shot tasks.
- The speedup for quantization and pruning should be evaluated. The performance improvement breakdown can help better understanding on those different compression methods for LLM.

**Questions For Authors:**

- Why do quantization first in the paper? How about pruning + quantization model?
- Is fine-tuning the key process for enhancing model performance?
- Why the quantized model with low rank can perform better than dense model in Table 6? Do the quantized models always show better accuracy across different tasks?
- It seems that unstructured pruning always performs better than 2:4 pruning in Table 7, but 2:4 pruned model has faster inference speed. Can you provide some insights on accuracy vs. inference speed of these different pruning methods?

**Relation To Broader Scientific Literature:**

LLM compression and combination of different compression works is efficient way for memory problem. Related works are referenced in the paper.

**Theoretical Claims:**

Yes. The SILM-Quant and SILM-LoRA algorithms were checked.

---

> ### Author Rebuttal · Authors · 2025-04-01
>
> We appreciate the reviewer's constructive feedback. Below, we provide answers to all the points raised.
>
> # Speedup Comparison without LoRA
>
> As requested, we have added two tables showing the layer-wise speedups of compressed models, with and without low-rank adapters, on an RTX-3060 GPU. While low-rank adapters slightly reduce speedup, the impact is minimal due to their low overhead. Importantly, SLiM-LoRA enables a flexible trade-off between speedup and model accuracy.
>
>
> \\begin{array}{|c|c|c|c|c|c|}
> \\hline
> \\rowcolor[gray]{0.9} Model Size & Batch Size & LoRA Type & Self-Attention & Up-Projection & Down Projection \\\\
> \\hline
> \\rowcolor[gray]{1.0} 13B & 16 & No LoRA & 3.89 & 3.86 & 3.86 \\\\
> \\rowcolor[gray]{0.95} 13B & 16 & FP16 & 2.18 & 2.53 & 2.60 \\\\
> \\rowcolor[gray]{1.0} 13B & 16 & INT4 & 1.28 & 3.24 & 3.17 \\\\
> \\hline
> \\rowcolor[gray]{0.95} 13B & 32 & No LoRA & 2.78 & 3.14 & 3.50 \\\\
> \\rowcolor[gray]{1.0} 13B & 32 & FP16 & 2.23 & 2.68 & 2.91 \\\\
> \\rowcolor[gray]{0.95} 13B & 32 & INT4 & 1.43 & 2.96 & 3.20 \\\\
> \\hline
> \\rowcolor[gray]{1.0} 13B & 64 & No LoRA & 1.46 & 1.88 & 1.98 \\\\
> \\rowcolor[gray]{0.95} 13B & 64 & FP16 & 1.38 & 1.78 & 1.67 \\\\
> \\rowcolor[gray]{1.0} 13B & 64 & INT4 & 1.21 & 1.69 & 1.65 \\\\
> \\hline
> \\end{array}
>
> \\begin{array}{|c|c|c|c|c|c|}
> \\hline
> \\rowcolor[gray]{0.9} Model Size & Batch Size & LoRA Type & Self-Attention & Up-Projection & Down Projection \\\\
> \\hline
> \\rowcolor[gray]{0.95} 70B & 16 & No LoRA & 3.63 & 4.22 & 4.03 \\\\
> \\rowcolor[gray]{1.0} 70B & 16 & FP16 & 2.18 & 2.86 & 2.75 \\\\
> \\rowcolor[gray]{0.95} 70B & 16 & INT4 & 3.11 & 3.99 & 3.79 \\\\
> \\hline
> \\rowcolor[gray]{1.0} 70B & 32 & No LoRA & 2.91 & 3.52 & 3.55 \\\\
> \\rowcolor[gray]{0.95} 70B & 32 & FP16 & 2.00 & 2.63 & 2.67 \\\\
> \\rowcolor[gray]{1.0} 70B & 32 & INT4 & 2.75 & 3.19 & 3.39 \\\\
> \\hline
> \\rowcolor[gray]{0.95} 70B & 64 & No LoRA & 1.89 & 1.98 & 2.12 \\\\
> \\rowcolor[gray]{1.0} 70B & 64 & FP16 & 1.38 & 1.70 & 1.86 \\\\
> \\rowcolor[gray]{0.95} 70B & 64 & INT4 & 1.51 & 1.77 & 1.94 \\\\
> \\hline
> \\end{array}
>
>
> # Accuracy Comparison with Dense Models
>
> We believe that **comparing a compressed model with a dense model of the same original parameter count can be misleading**. For gaining a better insight on the accuracy of the models, one needs to compare quality at iso-model size (e.g., effective number of parameters).
>
> `Figure-2-Page-8` presents the accuracy of different models vs. their parameter size in GB, allowing for a more fair comparison of the different models. Based on this figure, the **compressed models provide higher accuracies in comparison to dense models of the same parameter size (in GB)**. Additionally, SLiM improves the accuracy of the compressed models by adding negligible additional parameters to them. More discussions about this topic can be found in `Results-Section-Page-7` under the “Comparison of large compressed and small dense models” subsection.
>
> # Speedup Breakdown
>
> Please refer to our response to [Reviewer UKqS11](ADD-LINK-HERE) regarding an ablation study on the breakdown of the speedups.
>
> # Order of Pruning and Quantization
>
> Please see [our response to Reviewer SvGK14](https://openreview.net/forum?id=4UfRP8MopP&noteId=5SPU1BHLiL) for a detailed answer regarding this important question.
>
> # Effects of Fine-tuning
>
> SLiM is designed as a one-shot compression method (similar to SparseGPT) that delivers strong performance without any additional fine-tuning (`Table-1-Page-7`).
>
> While an **optional** PEFT step can further enhance accuracy, it is not central to SLiM’s approach. Improvements with optional PEFT are in `Table-2-Page-7` and `Table-7-Page-14`.
>
> # Improved Accuracy of Quantized Models
>
> With 4-bit quantization-only methods, performance is generally on par with dense models of the same effective size. As shown in `Table-6-Page-13`, some quantized models even slightly outperform their dense counterparts—a trend also observed in LQER [1] and QUIP# [2]. In such close cases, perplexity (`Table-9-Page-17`) can provide a more sensitive comparison metric.
>
>
> # Unstructured vs. 2:4 Sparsity
>
> Unstructured sparsity offers greater flexibility and often better accuracy, but is difficult to accelerate on modern GPUs [3, 4]. In contrast, 2:4 semi-structured sparsity is supported by recent GPUs (starting with Ampere architecture), enabling real speedups at the cost of some accuracy degradation.
>
> [1] Zhang, et al. “LQER: Low-Rank Quantization Error Reconstruction for LLMs”, ICML 2024
>
> [2] Tseng, et al. “QuIP#: Even Better LLM Quantization with Hadamard Incoherence and Lattice Codebooks”, ICML 2024
>
> [3] Xia, et al. “Flash-LLM: Enabling Cost-Effective and Highly-Efficient Large Generative Model Inference with Unstructured Sparsity”, VLDB 2023
>
> [4] Zheng, et al. “SparTA: Deep-Learning Model Sparsity via Tensor-with-Sparsity-Attribute”, OSDI 2022

---

### Official Review · Reviewer_SvGK · 2025-03-14

**Overall Recommendation:** 2

**Summary:**

The paper introduces SLIM, a one-shot post-training compression framework for large language models. It integrates three components: (1) quantization, (2) pruning for hardware‑friendly sparsity, and (3) a low‑rank adapter to compensate quantization errors. Experimental results show that SLIM improves model accuracy by up to approximately 5.66% and delivers significant GPU inference speedups, making it an effective solution for deploying large models in resource‑constrained environments.


**update after rebuttal**

I appreciate the authors' clarifications and the additional experimental results.

From my initial review through to the current rebuttal phase, my primary concern has consistently been the authors’ decision to minimize quantization error rather than directly targeting output error. In the first-round rebuttal, the authors did not provide a convincing justification for this design choice. In my reply-to-rebuttal comment, I reiterated this concern and requested a more thorough comparison between the two approaches.

In their latest response, the authors presented new experimental results aimed at addressing this issue. However, these results indicate that minimizing output error actually yields better accuracy than minimizing weight error. This directly contradicts the rationale for their initially chosen approach and undermines the justification for the proposed methodology. Based on this evidence, I remain unconvinced that the current approach is rationable.

I believe the methodology presented in this paper requires substantial revision and clearer justification of key design choices. While the paper has certain merits, I do not think it is ready for publication in its current form. I insist on a weak reject rating for this paper.

**Claims And Evidence:**

The paper provides extensive experimental results that support its claims of improved accuracy and significant inference speedups on GPUs.

**Essential References Not Discussed:**

Some recently published LLM quantization methods have been omitted from the discussion, such as:
[1] Egiazarian, V., Panferov, A., Kuznedelev, D., Frantar, E., Babenko, A., & Alistarh, D. (2024). Extreme compression of large language models via additive quantization. in ICML 2024.
[2] Tseng, A., Chee, J., Sun, Q., Kuleshov, V., & De Sa, C. (2024). Quip#: Even better llm quantization with hadamard incoherence and lattice codebooks. in ICML 2024.

**Experimental Designs Or Analyses:**

The experimental designs and analyses are generally sound, with extensive evaluations on standard benchmarks and comparisons against SOTA methods on popular model families (LLaMA‑2, OPT). However, there are some minor concerns. For example, while the experiments validate overall accuracy improvements and speedup claims, the lack of detailed ablation studies—particularly regarding the ordering of quantization and pruning—limits our understanding of error propagation in the cascaded approach. Moreover, the trade-off between minimizing quantization error versus feature or activation error is not thoroughly explored, which could affect the validity of the conclusions drawn from the reported experiments.

**Methods And Evaluation Criteria:**

The proposed methods and evaluation criteria are generally well-aligned with the goal of compressing large language models. However, certain design choices are less convincing. For instance, the decision to minimize quantization error rather than directly targeting activation or output error is not thoroughly justified, and the cascaded application of quantization followed by off-the-shelf pruning raises concerns about potential error amplification. Without sufficient ablation studies or evidence that this ordering is optimal, these aspects leave room for skepticism regarding the overall effectiveness of the approach.

**Other Comments Or Suggestions:**

None.

**Other Strengths And Weaknesses:**

**Strengths:**
- The paper creatively combines probabilistic quantization, off‑the‑shelf pruning, and a saliency‑based low‑rank adapter into a unified one‑shot compression framework.
- By targeting hardware‑friendly sparsity (e.g., 2:4 patterns) and demonstrating significant inference speedups on GPUs, the approach is well-suited for real-world deployment of large language models.

**Weaknesses:**
- The decision to minimize quantization error instead of directly targeting activation or output error is not thoroughly justified, potentially limiting overall performance.
- The cascaded application of quantization followed by off‑the‑shelf pruning risks compounding errors, with no sufficient justification provided for this specific ordering.
- There is a lack of joint optimization between quantization and pruning, which may exacerbate error accumulation.
- The theoretical derivations rely on assumptions about weight distributions and saliency properties that are not fully validated, potentially affecting the robustness of the method.

**Questions For Authors:**

1. Can you elaborate on why you chose to minimize quantization error rather than targeting activation or output error? Have you performed any experiments to compare the two approaches? Notably, recent methods such as QUIP# and AQLM focus on minimizing activation error for each layer or group.

2. What is the rationale for applying quantization before pruning? Did you consider or experiment with reversing the order, and if so, what were the outcomes?

3. Is it possible to integrate quantization and pruning into a joint optimization framework rather than treating them as separate cascaded steps? If so, what are the trade-offs, and how do you justify not pursuing this integrated approach?

The justification for minimizing quantization error instead of directly targeting activation or output error, as well as the possibility of integrating quantization and pruning into a joint optimization framework, are my two major concerns. I may reconsider my rating based on the authors' responses to these questions.

**Relation To Broader Scientific Literature:**

The paper’s contributions build directly on established work in model quantization, and pruning. It extends prior methods (post-training quantization approaches and pruning techniques like SparseGPT or Wanda), by proposing SLIM‑Quant, a probabilistic formulation for uniform quantization that reframes error minimization as a convex problem via numerical integration. Additionally, it integrates ideas from low‑rank adaptation research ( like L2QER and LoRA) by introducing a saliency‑based low‑rank adapter (SLIM‑LoRA) that leverages invertible and additive saliency functions to compensate for compression errors . By combining these approaches into a unified, cascaded pipeline, the work aims to deliver efficient compression and hardware‑friendly inference for LLM.

**Theoretical Claims:**

I reviewed the derivations for the probabilistic formulation of quantization error minimization (Equations 3–7) as well as the formulation of the saliency-based low-rank adapter (Equations 8–11). While the derivations are logically coherent, they depend on assumptions that are not rigorously justified. For example, assumptions about the underlying weight distribution for numerical integration and the properties (invertibility and additivity) of the proposed saliency function.

---

> ### Author Rebuttal · Authors · 2025-04-01
>
> # Minimizing Quantization Error
>
> We agree that minimizing the final output error, i.e., $|XW - XW^C|$, is the ideal objective for any compression method. However, directly optimizing this quantity is computationally intractable in general. It is known to be NP-Hard and difficult to scale across layers.
>
> In SLiM, we address this challenge by decomposing the problem into two tractable subgoals: (1) minimizing weight quantization error in a close-form, and (2) applying a lightweight, saliency-guided LoRA module to recover the residual output error post quantization. This design choice allows SLiM to scale to large models, while achieving state-of-the-art results across multiple tasks. Our ablation studies (`Table-1,6`) support the effectiveness of this decomposition: using LoRA on top of weight quantization consistently improves output accuracy, validating that our approach approximates the harder output-error objective effectively in practice.
>
> # Comparison with QUIP# and AQLM
>
> We compare SLiM’s quantization-only (no pruning) zero-shot accuracy on LLaMA-2 models against QUIP# and AQLM. QUIP# does not support sparsity due to its Hadamard transform densifying sparse weights, and AQLM quantization of sparse models takes days (as reported on their code base). We are processing these with AQLM and will share results promptly, given rebuttal constraints.  SLiM outperforms other methods in 2 out of 4 tasks..
>
> ## LLaMA-2 7B (4-bit Quantization)
>
> \\begin{array}{|c|c|c|c|c|}
> \\hline
> \\rowcolor[gray]{0.9} \\textbf{Method} & \\textbf{Arc Challenge} & \\textbf{Arc Easy} & \\textbf{PiQA} & \\textbf{Winogrande} \\\\
> \\hline
> \\rowcolor[gray]{0.95} QUIP-Sharp & 40.5 & \\textbf{69.1} & \\textbf{78.4} & 67.6 \\\\
> \\rowcolor[gray]{1.0} AQLM & 40.3 & 68.9 & 77.7 & 67.3 \\\\
> \\rowcolor[gray]{0.95} SLiM-Quant + SLiM-LoRA & \\textbf{43.8} & 68.4 & 78.1 & \\textbf{68.4} \\\\
> \\hline
> \rowcolor[gray]{0.9} \\Delta \text{with best alternative method} & +3.3 & -0.7 & -0.3 & +0.8 \\\\
> \\hline
> \\end{array}
>
> ## LLaMA-2 13B (4-bit Quantization)
>
> \\begin{array}{|c|c|c|c|c|}
> \\hline
> \\rowcolor[gray]{0.9} \\textbf{Method} & \\textbf{Arc Challenge} & \\textbf{Arc Easy} & \\textbf{PiQA} & \\textbf{Winogrande} \\\\
> \\hline
> \\rowcolor[gray]{0.95} QUIP-Sharp & 45.5 & \\textbf{73.9} & \\textbf{78.9} & 69.9 \\\\
> \\rowcolor[gray]{1.0} AQLM & 43.9 & 72.2 & 78.6 & 70.4 \\\\
> \\rowcolor[gray]{0.95} SLiM-Quant + SLiM-LoRA & \\textbf{47.1} & 72.5 & 78.5 & \\textbf{72.5} \\\\
> \\hline
> \rowcolor[gray]{0.9} \\Delta \text{with best alternative method} & +1.6 & -1.4 & -0.4 & +2.1 \\\\
> \\hline
> \\end{array}
>
> # Order of Pruning and Quantization
>
> We evaluated two SLiM variants: (1) Prune-First, where pruning is applied before quantization, and (2) Quantize-First, where quantization precedes pruning. The table below reports average accuracy across six zero-shot tasks and shows that the compression order has a negligible effect on performance. In both cases, SLiM-LoRA effectively mitigates any induced errors.
>
>
> \\begin{array}{|c|c|c|c|c|c|c|c|c|c|}
> \\hline
> \\rowcolor[gray]{0.9}Method&Structure&OPT125M&OPT350M&OPT1.3B&OPT2.7B&OPT6.7B& OPT13B & LLaMA2-7B & LLaMA2-13B  \\\\
> \\rowcolor[gray]{0.95} Quantize First & 2:4 & 34.62 & \\textbf{34.36} & 40.61 & \\textbf{42.73} & 45.99 & \\textbf{46.09} & 51.15 & \\textbf{54.94} \\\\
> \\rowcolor[gray]{1.0} Prune First & 2:4 & \\textbf{34.81} & 33.80 & \\textbf{40.66} & 42.10 & \\textbf{46.02} & 45.15 & \\textbf{51.50} & 54.77 \\\\
> \\hline
> \\rowcolor[gray]{0.95} Quantize First & Unstructured & 35.20 & \\textbf{35.32} & 41.85 & \\textbf{43.48} & 47.08 & \\textbf{47.96} & 54.26 & \\textbf{57.85}  \\\\
> \\rowcolor[gray]{1.0} Prune First & Unstructured & \\textbf{35.46} & 35.06 & \\textbf{41.49} & 43.16 & \\textbf{47.09} & 46.87 & \\textbf{53.61} & 57.94 \\\\
> \\hline
> \\end{array}
> # Joint Quantization and Pruning
> Thank you for the suggestion. With the current formulation of SLiM, joint optimization of pruning and quantization is not feasible. However, because of the SLiM’s unique decomposition of tasks, SLiM-LoRA  is compatible with various compression methods and one can readily use it regardless of the compression order.
> # Assumptions in Theoretical Derivations
> - **SLiM-Quant:** For SLiM-Quant, we avoid assumptions about the weight matrices’ distribution during optimization, instead using an empirically derived histogram of the weights to guide the process.
> - **SLiM-LoRA:** For SLiM-LoRA’s saliency function, we assume it is additive and invertible.
>    - **Additivity** holds as we define $F(M)=diag(x)M$, where $x$ is the layer’s average input. For matrices $A$ and $B$, $F(A+B)=diag(x)(A+B)=diag(x)A+diag(x)B=F(A)+F(B)$.
>    - **Invertibility** is ensured by guaranteeing that $diag(x)$ is non-singular, achieved by enforcing positive values in $x$ (see line 5, Algorithm 2, page 5). This allows the inverse to be computed as $F^{-1}(M) = diag(1/x)M$. These properties enable SLiM-LoRA to effectively map recovered saliency to low-rank adapters.

---

> > ### Comment · Reviewer_SvGK · 2025-04-03
> >
> > Thanks for the clarifications, but my concerns remain insufficiently addressed.
> >
> > While I appreciate the discussion regarding computational intractability, many existing methods demonstrate that approximate or layer-wise strategies for minimizing output error can be effective. Merely stating that the ideal objective is NP-hard does not justify choosing weight-error minimization by default, particularly when practical approximations exist. Even if SLiM outperforms QUIP# and AQLM in some cases, the argument lacks persuasiveness unless there is a direct comparison between weight-error minimization and output-error minimization under the same framework of SLiM.
> >
> > Furthermore, the response asserts that joint optimization is not feasible within SLIM’s modular design but fails to provide clear experimental or theoretical evidence to confirm that it cannot be implemented effectively or would not yield superior results.
> >
> > These issues continue to raise concerns regarding the overall design and justification of the approach.

---

> > > ### Author Response · Authors · 2025-04-07
> > >
> > > Thank you for reviewing our response and raising additional valuable points. In accordance with your suggestions, we have further extended our method to address your concerns.
> > >
> > > ---
> > >
> > > # Output Error Minimization in SLiM-Quant
> > >
> > > We extended SLiM-Quant by incorporating an output error minimization approach inspired by [AWQ [1]](https://arxiv.org/abs/2306.00978). Similar to AWQ, our revised algorithm applies a scaling strategy to activations, reducing the quantization error of salient weight channels. Specifically, we scale up the weights associated with the most significant channels and correspondingly scale down the related input activations. This approach maintains computational equivalence while effectively lowering the quantization-induced output error. Notably, scaling approximately 1% of the channels does not alter the overall quantization parameters but significantly reduces errors in the critical channels.
> > >
> > > However, our approach diverges from AWQ by introducing a novel saliency metric that jointly considers both activations and weights. We define the saliency of each channel as the product of the normalized average magnitudes of inputs and weights, expressed as ${|x|} \odot {|w|}$, where ${|x|}$ and ${|w|}$ denote the average magnitudes of activations and weights, respectively. Channels with the highest saliency are scaled by a factor $s > 1$, while their corresponding activations are scaled by $\frac{1}{s}$. Although this method introduces modest computational overhead, that is attributed to on-the-fly adjustments of roughly 1% of activations and resulting irregular memory access patterns, it yields measurable accuracy improvements.
> > > These results underscore a clear trade-off between computational complexity and model performance, highlighting the relative strengths of SLiM$^{O}$ (SLiM with output error minimization) over SLiM$^{W}$ (SLiM with weight error minimization).
> > >
> > > ## Average Accuracy Over 6 Zero-shot Tasks
> > >
> > > ### 2:4 Sparsity
> > >
> > > \\begin{array}{|c|c|c|}
> > > \\hline
> > > \\rowcolor[gray]{0.9} \\textbf{Method} & \\textbf{LLaMA-2-7B} & \\textbf{LLaMA-2-13B} \\\\
> > > \\hline
> > > \\rowcolor[gray]{0.95} SLiM^{{W}} & 51.15 & 54.94 \\\\
> > > \\rowcolor[gray]{1.0} SLiM^{{O}} & 51.22 & 55.05 \\\\
> > > \\hline
> > > \\end{array}
> > >
> > >
> > > ## Unstructured Sparsity
> > >
> > > \\begin{array}{|c|c|c|}
> > > \\hline
> > > \\rowcolor[gray]{0.9} \\textbf{Method} & \\textbf{LLaMA-2-7B} & \\textbf{LLaMA-2-13B} \\\\
> > > \\hline
> > > \\rowcolor[gray]{0.95} SLiM^{{W}} & 54.26 & 57.85 \\\\
> > > \\rowcolor[gray]{1.0} SLiM^{{O}} & 54.46 & 57.97 \\\\
> > > \\hline
> > > \\end{array}
> > >
> > > ## Perplexity on WikiText-2
> > >
> > > ### 2:4 Sparsity
> > > \\begin{array}{|c|c|c|}
> > > \\hline
> > > \\rowcolor[gray]{0.9} \\textbf{Method} & \\textbf{LLaMA-2-7B} & \\textbf{LLaMA-2-13B} \\\\
> > > \\hline
> > > \\rowcolor[gray]{0.95} SLiM^{{W}} & 7.56 & 6.50 \\\\
> > > \\rowcolor[gray]{1.0} SLiM^{{O}} & 7.35 & 6.38 \\\\
> > > \\hline
> > > \\end{array}
> > >
> > >
> > > ## Unstructured Sparsity
> > > \\begin{array}{|c|c|c|}
> > > \\hline
> > > \\rowcolor[gray]{0.9} \\textbf{Method} & \\textbf{LLaMA-2-7B} & \\textbf{LLaMA-2-13B} \\\\
> > > \\hline
> > > \\rowcolor[gray]{0.95} SLiM^{{W}} & 6.16 & 5.36 \\\\
> > > \\rowcolor[gray]{1.0} SLiM^{{O}} & 6.06 & 5.28 \\\\
> > > \\hline
> > > \\end{array}
> > >
> > > ---
> > >
> > > # Joint Pruning and Quantization
> > >
> > > The current modular design of SLiM does not support direct joint pruning and quantization. However, as noted in our rebuttal, our proposed saliency-based LoRA method (SLiM-LoRA) is compatible with existing joint pruning and quantization approaches. To demonstrate this compatibility, we integrated SLiM-LoRA with JSQ [2], a representative joining pruning and quantization method. The table below reports the average accuracy across six zero-shot tasks for different models:
> > >
> > > \\begin{array}{|c|c|c|c|c|}
> > > \\hline
> > > \\rowcolor[gray]{0.9} \\textbf{Method} & \\textbf{LoRA} & \\textbf{Structure} & \\textbf{LLaMA-2-7B} & \\textbf{LLaMA-2-13B} \\\\
> > > \\hline
> > > \\rowcolor[gray]{0.95} \\text{JSQ (4-bit)} & \\text{N/A} & 2{:}4 & 45.34 & 49.45 \\\\
> > > \\rowcolor[gray]{1.0} \\text{JSQ (4-bit)} & \\text{SLiM-LoRA} & 2{:}4 & 46.14 & 50.19 \\\\
> > > \\rowcolor[gray]{0.95} \\text{JSQ (4-bit)} & \\text{N/A} & \\text{Unstructured} & 52.08 & 56.20 \\\\
> > > \\rowcolor[gray]{1.0} \\text{JSQ (4-bit)} & \\text{SLiM-LoRA} & \\text{Unstructured} & 52.37 & 56.72 \\\\
> > > \\hline
> > > \\end{array}
> > >
> > > These results demonstrate that applying SLiM-LoRA enhances the accuracy of models utilizing joint pruning and quantization.
> > >
> > > Please note that JSQ, even when augmented with SLiM-LoRA, does not outperform SLiM-Quant combined with SLiM-LoRA. This discrepancy arises because JSQ was originally designed for 8-bit weight quantization, whereas SLiM-Quant is specifically optimized for 4-bit weight quantization. To facilitate a fair comparison, we extended JSQ to support 4-bit weight quantization (see `Section-4-Page-6`, 'Baselines' for detailed explanations).
> > >
> > > [1] Lin, et al. “AWQ: Activation-aware Weight Quantization for LLM Compression and Acceleration”, MLSys 2024
> > >
> > > [2] Yu, et al. “JSQ: Compressing Large Language Models by Joint Sparsification and Quantization”, ICML 2024

---

### Decision · Program_Chairs · 2025-05-01

**Decision:**

Accept (poster)

**Comment:**

The paper introduces SLIM, a new one-shot LLM compression framework.  SLIM consists of quantization (via SLIM-Quant, a new probabilistic formulation of quantization error minimization), pruning (via off-the-shelf methods, e.g., Wanda), and low-rank adaptation (via SLIM-LoRA a novel saliency-based method, using calibration data).  Notably, to further reduce memory overhead of the adapters introduced in SLIM-LoRA, the paper also further quantizes the saliency-based adapters, leading to SLIM-LoRA^Q.  The efficacy of SLIM is compared to other SOTA one-shot post-training quantization (1shot-PTQ) schemes, i.e., SparseGPT, Wanda, L^{2}QER, JSQ, as well as a naive version of SLIM-LoRA (called Naive-LoRA).  SLIM outperforms previous 1shot-PTQ methods across a wide range of OPT/Llama-2 model sizes, average common sense reasoning benchmarks, and sparsity patterns (i.e., 2:4 and unstructured).  Impressive speedups are shown post-SLIM for both 16-bit and 4-bit precisions, as well as on commodity and high-end GPUs (the former are found in the Appendix).    The appendix contains a expansive range of experiments; due to the borderline nature of the review, I extensively consulted the main text/appendix as well as the source code to confirm lingering reviewer concerns were taken of.

The paper is well written and the 3 stages of SLIM are intuitive.  Many reviewers had follow up questions about the various stages, which the authors did well in answering by cross-referencing the results from the main paper.  However, to proactively address some of the possible confusing stages of the pipeline--e.g., SLIM-LoRA naturally leads readers to think of fine-tuning, but this is actually saliency-based one-shot calibration, which may be followed up by actual fine-tuning--the authors should add a summary of contributions at the end of the intro, including a delineation between the one-shot calibration stage in SLIM-LoRA versus actual fine-tuning (e.g., anticipating the second question from Reviewer q6Mp as a regular source of confusion when reading the paper and the authors reply as an early statement to clear this up).

Reviewer SvGK raised major concerns by the authors’ decision to minimize quantization error rather than directly targeting output error.  The authors addressed this concern in a variety of ways, resulting in further experiments comparing the two approaches: i) minimizing output error (MOE), and (ii) minimizing weight error (MWE).  Ultimately, MOE outperformed MWE, which Reviewer SvGK noted could not address his concerns.  However, the performance discrepancy is extremely small, which means that MWE achievers near MOE without MOE's computational difficulties (an inclusion of this discussion in the appendix will similarly address future readers' concerns).  Moreover, both Reviewer UKqS  and q6Mp note that the frameworks of MOE and MWE are ultimately heuristic design choices, and the version of MWE serving as the backbone for SLIM outperform existing non-SLIM SOTA methods.  Thus, SLIM beats SOTA (which ultimately matters for this paper), and the methodology of further improving SLIM by incorporating MOE may be addressed in future work.

Additional reviewer concerns include potential generalization issues stemming from the calibration set.  However, these were raised late in the discussion period and are extensively addressed in Figure 4 and Table 18 of the appendix.  Reviewing all the submitted materials (main text, appendices, entire codebase), I appreciate the lengths the authors took to recreate the results of the previous SparseGPT and Wanda papers (e.g., using the same C4 sampling and calibration set curation from SparseGPT).  This is a point that should be stated clearly (and proudly) in the main paper (as it was in the Wanda paper, Section 4 third paragraph), which can quickly allay reader questions about comparison-fairness in an information-dense paper such as this.  I also would have liked to see more Llama-3 results in the main text (and it is worth pointing out those in the appendix), although it is understandable that the authors reproduced previous SOTA methods using their supplied code built for Llama-2 and OPT.